# Revealing Vision-Language Integration in the Brain with Multimodal Networks

## Abstract

We use multimodal deep neural networks to identify sites of multimodal integration in the human brain. These are regions where a multimodal language-vision model is better at predicting neural recordings (stereoelectroencephalography, SEEG) than either a unimodal language, unimodal vision model, or a linearly-integrated language-vision model. We use a wide range of state-of-the-art models spanning different architectures including Transformers and CNNs (ALBEF, BLIP, Flava, ConvNeXt, BEIT, SIMCLR, CLIP, SLIP) with different multimodal integration approaches to model the SEEG signal while subjects watched movies. As a key enabling step, we first demonstrate that the approach has the resolution to distinguish trained from randomly-initialized models for both language and vision; the inability to do so would fundamentally hinder further analysis. We show that trained models systematically outperform randomly initialized models in their ability to predict the SEEG signal. We then compare unimodal and multimodal models against one another. A key contribution is standardizing the methodology for doing so while carefully avoiding statistical artifacts. Since models all have different architectures, number of parameters, and training sets which can obscure the results, we then carry out a test between two controlled models: SLIP-Combo and SLIP-SimCLR which keep all of these attributes the same aside from multimodal input. Using this method, we identify neural sites (on average 141 out of 1090 total sites or 12.94%) and brain regions where multimodal integration is occurring. We find numerous new sites of multimodal integration, many of which lie around the temporoparietal junction, long theorized to be a hub of multimodal integration.

## 1 Introduction

We expand the use of deep neural networks for understanding the brain from unimodal models, which can be used to investigate language and vision regions in isolation, to multimodal models, which can be used to investigate language-vision integration. Beginning with work in the primate ventral visual stream (Yamins et al., 2014; Schrimpf et al., 2020), this practice now includes the study of both the human vision and language cortex alike (Chang et al., 2019; Allen et al., 2021; Bhattasali et al., 2020; Nastase et al., 2021; Schrimpf et al., 2021; Goldstein et al., 2021; 2022; Lindsay, 2021; Caucheteux & King, 2022; Conwell et al., 2022). These studies, however, focus on a single modality of input – vision alone or language alone. Yet, much of what humans do fundamentally requires multimodal integration.

As a product of this unimodal focus, we have learned far less about the correspondence between biological and artificial neural systems tasked with processing visual and linguistic input *simultaneously*. Here, we seek to address this gap by using performant, pretrained multimodal deep neural network (DNN) models (ALBEF, BLIP, Flava, SBERT, BEIT, SimCSE, SIMCLR, CLIP, SLIP) (Li et al., 2021; 2022b; Singh et al., 2022; Bao et al., 2021; Gao et al., 2021; Chen et al., 2020; Radford et al., 2021; Mu et al., 2021) to predict neural activity in a large-scale stereoelectroencephalography (SEEG) dataset consisting of neural responses to the images and scripts of popular movies (Yaari et al., 2022) collected from intracranial electrodes. The goal is to use systematic comparisons between the neural predictivity of unimodal and multimodal models to identify sites of vision-language integration in the brain.

We make the following contributions:

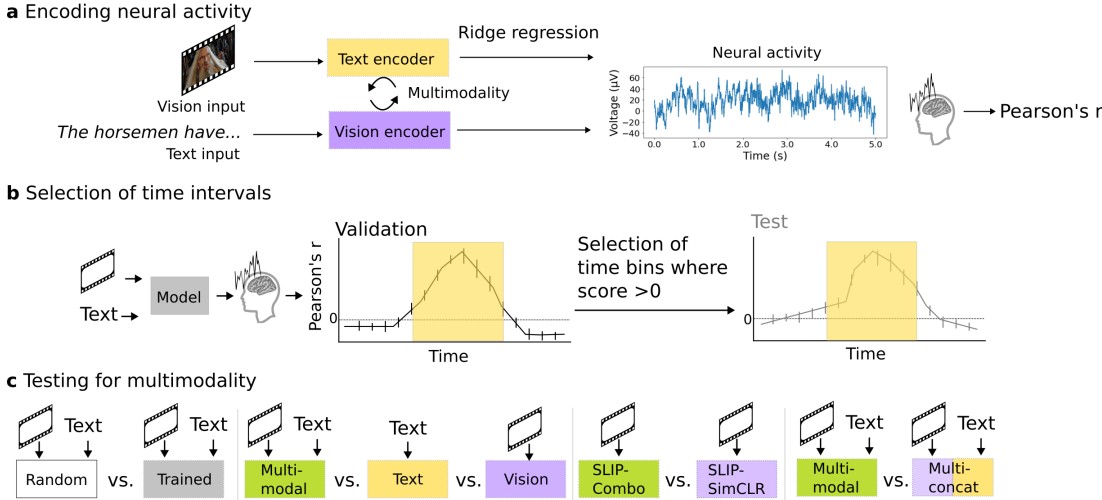

Figure 1: (a) We parse the stimuli, movies, into image-text pairs (which we call *event structures*) and process these with either a vision model, text model, or multimodal model. We extract feature vectors from these models and predict neural activity in 161 25ms time bins per electrode, obtaining a Pearson correlation coefficient per time bin per electrode per model. We run this regression using both trained and randomly initialized encoders and for two datasets, a vision-aligned dataset and language-aligned dataset, which differ in the methods to sample these pairs. (b) We design a bootstrapping test over input image-text pairs to build 95% confidence intervals on scores per time bin per electrode. We filter out time bins in electrodes where the validation lower confidence interval is less than zero. (c) The first analysis of this data investigates if trained models outperform randomly initialized models. The second analysis investigates if multimodal models outperform unimodal models. The third analysis repeats the second holding constant the architecture and dataset to factor out these confounds. Two other analyses are described in the text. The fourth analysis investigates if multimodal models outperform models that concatenate language and vision features.

1. We introduce a statistically-rigorous methodology to compare multimodal models against neural data, against one another, and against unimodal models. We release a code toolbox to perform this analysis and enable future work.
2. We demonstrate that this method is sufficiently fine-grained to distinguish randomly initialized from trained vision, language, and multimodal models. Previously this had been questionable for language models and never investigated for multimodal models. Without this gap, we could not conclude that multimodal processing is taking place, merely that multimodal architectures are generically helpful.
3. Using a wide array of models, we employ this method to identify areas associated with multimodal processing, i.e. areas where multimodal models outperform unimodal models as well as language-vision models with linearly-integrated features e.g. concatenation of vision and language features.
4. We then introduce an architecture-, parameter-, and dataset-controlled experiment where two variants of the same model, one unimodal and the other multimodal, are used to identify multimodal regions.
5. We catalogue a collection of areas which perform multimodal integration and are candidates for future experiments along with the time-course of that integration. We also catalogue the best candidate multimodal models that best explain activity in the areas associated with multimodal integration.

These experiments for the first time connect multimodal networks and multimodal regions in the brain, significantly expanding the range of questions and brain areas that can be investigated with deep neural networks in the brain.

## 2 RELATED WORK

Multimodal vision and language processing in the brain is presumed to show some degree of localization based on neuroscience experiments where subjects are presented with specially constructed multimodal visio-linguistic stimuli and the response is measured using functional Magnetic Reso-

nance Imaging (fMRI) against several control stimuli. For example, recent multivariate fMRI studies have identified the superior temporal cortex to be associated with specific forms of auditory and visual processing (Jouen et al., 2015; Zhang et al., 2023; Van Audenhaege et al., 2023; Friederici et al., 2009; Friederici, 2012). Furthermore, deeper semantic integration of vision and language has been seen in the middle temporal and inferior parietal cortex (Petrides, 2023; Bzdok et al., 2016). Other areas include the supramarginal gyrus, involved in emotion processing (Stoeckel et al., 2009), the superior frontal lobe, commonly associated with self-awareness (Schilling et al., 2013), the caudal middle frontal cortex, commonly associated with eye movements and scene understanding (Badre & Nee, 2018), and the pars orbitalis, which contains Broca's area and is associated with speech processing (Belyk et al., 2017).

There has been considerable interest in investigating the effectiveness of representations from neural networks in modeling brain activity (Wehbe et al., 2014; Kuzovkin et al., 2018; Conwell et al., 2021; Goldstein et al., 2021; 2022; Lindsay, 2021; Caucheteux & King, 2022). These approaches have typically employed various forms of linear regressions to predict brain activity from the internal states of candidate models, with specific modifications to the prediction depending on the type of neural recording used. The majority of these works tend to focus on vision or language alone, in large part because unimodal datasets (Chang et al., 2019; Allen et al., 2021; Bhattasali et al., 2020; Nastase et al., 2021) and unimodal models (e.g. PyTorch-Image-Models; Huggingface) are the most commonly available. Prior experiments have shown that language-based unimodal networks and vision-based unimodal networks effectively model activity in the brain. Many prior experiments include comparisons between trained and randomly initialized networks and have shown that trained unimodal vision networks model activity better than randomly initialized networks but have struggled to reproduce the same result in language networks (Schrimpf et al., 2021).

More recent papers have emerged demonstrating the ability for multimodal networks to predict fMRI activity in the higher visual cortex in response to visual stimuli better than unimodal vision models (Wang et al., 2022; Oota et al., 2022), indicating that multimodal networks are integrating deeper semantic information similar to the brain. Our work differs from this and other prior work by considering general vision-language integration. We employ multimodal networks including ALBEF, CLIP, and SLIP and use representations from these networks to predict brain activity up to 2000ms before and after the occurrence of an event. Our results unveil a number of distinct electrodes wherein the activity from multimodal networks predicts activity better than any unimodal network, in ways that control for differences in architecture, training dataset, and integration style where possible. In contrast to most previous work that mainly leverages fMRI, here we focus on high-fidelity neurophysiological signals similar to Kuzovkin et al. (2018) We use this analysis to identify sites of mutlimodal integration, many of which align with and overlap with areas mentioned in prior work. Our work is among the first to establish these sites of integration and present which network is the best at predicting activity in these sites.

## 3 METHODS

**Neural Data:** Invasive intracranial field potential recordings were collected during 7 sessions from 7 subjects (4 male, 3 female; aged $4 - 19$, $\mu = 11.6$, $\sigma = 4.6$) with pharmacologically intractable epilepsy. During each session, subjects watched a feature length movie from the Aligned Multimodal Movie Treebank (AMMT) (Yaari et al., 2022) in a quiet room while neural activity was recorded using SEEG electrodes (Liu et al., 2009) at a rate of 2kHz.

We parse the neural activity and movie into language-aligned events (word onset) and visually-aligned events (scene cuts) where each event consists of an individual image-text pair and create two stimulus alignment datasets where we have coregistered visual and language inputs to the given models. The language-aligned dataset consists of word utterances with their sentence context for text input and the corresponding closest frame to the word onset. The vision-aligned dataset consists of scene cuts and the closest sentence to occur after the cut. Word-onset times are collected as part of the AMMT metadata and visual scene cuts are extracted from each movie using PySceneDetect (Castellano, 2022). Following Goldstein et al. (2021), we extract a 4000ms window of activity (about 8000 samples), 2000ms prior to the event occurrence and 2000ms after the event occurrence, per electrode. We split the 4000ms window into sub-windows of 200ms with a sliding window of 25ms and the activity is averaged per sub-window to get a series of mean activity values over time per electrode. Further details of the neural data processing can be found in Appendix A.

**Models**: We use 12 pretrained deep neural network models, 7 multimodal and 5 unimodal, to explore the effect of multimodality on predictions of neural activity. The models that serve as our main experimental contrast are the SLIP models (Mu et al., 2021). The SLIP models are a series of 3 models that use the same architecture (ViT-[S,B,L]) and the same training dataset (YFCC15M), but are trained with one of three objective functions: pure unimodal SimCLR-style (Chen et al., 2020) visual contrastive learning (henceforth SLIP-SimCLR), pure multimodal CLIP-style (Radford et al., 2021) vision-language alignment (henceforth SLIP-CLIP), and combined visual contrastive learning with multimodal CLIP-style vision-language alignment (henceforth SLIP-Combo). The full set constitutes a set of 5 models (SLIP-SimCLR; the SLIP-CLIP visual encoder; the SLIP-CLIP language encoder; the SLIP-Combo visual encoder; the SLIP-Combo language encoder). For more general (uncontrolled) multimodal-unimodal contrasts, we include multimodal models ALBEF (Li et al., 2021), BLIP (Li et al., 2022b), and Flava (Singh et al., 2022) and the unimodal models SBERT (Reimers & Gurevych, 2019b), BEIT (Bao et al., 2021), ConvNeXt (Liu et al., 2022b), and SimCSE (Gao et al., 2021). Furthermore, as part of our initial multimodal set, we introduce *linearly-integrated language-vision networks*. Our first model, *MultiConcat* concatenates representations from a pretrained SimCSE and pretrained SLIP-SimCLR. Our second model, *MultiLin* performs the same concatenation and trains a linear projection using the NLVR-2(Suhr et al., 2018) dataset. For each of the 14 networks, we assess both a pretrained and randomly-initialized version to assess whether the multimodality we assume in the brain coincides with features learned in training the multimodal models. This gives us a total of 28 networks for analysis. Random initialization of these networks has different effects on the multimodal status of particular networks. Since SLIP-Combo and SLIP-CLIP are designed to be multimodal due to contrastive training, randomly initialized SLIP-Combo or SLIP-CLIP are considered unimodal in the analysis. The multimodal signal used to guide model predictions is lost due to the random initialization in this case. However, for ALBEF, BLIP, and Flava, these are architecturally multimodal models that directly take both modalities as input regardless of random initialization. Random initialization for these three networks has no effect on the multimodal status of output representations. Details on the logic behind these choices are given in Appendix B.

**Neural Regression:** To identify multimodal electrodes and regions in the brain, we first extract feature vectors from every layer of the candidate networks using the image-text pairs in a given dataset alignment. We then use these features from each layer as predictors in a 5-fold ridge regression predicting the averaged neural activity of a target neural site in response to each *event structure* (defined here as an image-text pair). Per fold, we split our dataset of event structures contiguously based on occurrence in the stimulus. We place 80% of the event structures in the training set, 10% of event structures in the validation set, and 10% in the testing set. We use contiguous splitting to control for the autoregressive nature of the movie stimuli. We measure the strength of the regression using the Pearson correlation coefficient between predicted average activity and actual average activity for a *specific* time window in each neural site for a held-out test set of event structures. Two aspects of this process are worth emphasizing: First, the final performance metric (the Pearson correlation between actual and predicted neural activity for a held-out test set of event-structures) is not a correlation over time-series (for which the Pearson correlation is inappropriate), but a correlation over a set of (nominally IID) event-structures that we have extracted by design to minimize the autoregressive confounds of time-series data. Second, the cross-validation procedure and train-test splitting is specifically designed to assess the generalization of the neural regression fits, and as such contains no cross-contamination of selection procedures (e.g. the maximally predictive layer from a candidate model, feature normalization, or the ridge regression lambda parameter) and final model scoring. In this case, we use the cross-validation procedure to select the scores associated with the best performing layer and select the best performing regression hyperparameters. Further details on the regression method can be seen in Appendix C.

**Bootstrapped Confidence Intervals across Time**: In order to make model comparisons on a sound statistical basis, we use a bootstrapping procedure over image-text pairs in a given dataset alignment to calculate 95% confidence intervals on the correlation scores per time bin for the training, validation, and test set alike.

Our bootstrapping procedure involves first resampling the image-text pairs and corresponding neural activity with replacement and then re-running the regression with the resampled event structures, predicting the associated neural activity per time bin per electrode. We run the resampling 1000 times and use the same resampled event structures across all models to allow for model comparison. Directly mimicking the standard encoding procedure, this bootstrapping leaves us with 95% confidence

intervals on the predictive accuracy of a given model per time bin per electrode across all of the training, validation, and test splits. We obtain two sets of confidence intervals per dataset alignment, either language-aligned or vision-aligned. In subsequent model comparisons, we use the 95% confidence interval over the validation set to filter out time bins per electrode in which either of the model's scores was not significantly above 0. Subsequent analysis uses the held-out test set scores for analysis.

**Model Comparisons**: Taking inspiration from fMRI searchlight analyses (Kriegeskorte et al., 2006; Etzel et al., 2013), we next perform a series of statistical tests on each electrode to determine whether or not they are better predicted by multimodal or unimodal representations and whether each electrode is better predicted by representations from trained models or randomly initialized models.

We first filter all time bins in electrodes for models where the lower 95% confidence interval of the validation score overlapped with zero. This ensures that the analysis focuses on time bins and electrodes where meaningful neural signal is occurring. We remove models from further analysis on a particular electrode if that model has a confidence interval that overlaps with zero for all time bins, on the validation set. If only one model has at least 10 time bins with this requirement (a minimal threshold for bootstrapped comparisons), we consider this model the best model by default and do no further processing on the electrode.

For electrodes without these "default winners", we employ an additional statistical test of model difference between the first and second-highest ranking models for a given comparison. That is, we use a second-order bootstrapping procedure (this time across time bins, rather than across event structures), calculating the difference in the average score across resampled time bins between the 2 candidate models in a given comparison. This procedure is designed to minimize the possibility of one model producing a random peak of predictivity that does not adequately reflect its predictivity more generally, and may artificially give the impression of being the superior model in a comparison. We run this for model pairs on electrodes that have at least 10 time bins remaining after filtering based on the lower confidence interval of the validation set for both models. For the bootstrapping procedure of model difference, we identify electrodes where the difference in performance is statistically significant and use FDR (Benjamni-Hochberg) multiple comparisons corrections to adjust the p-value associated with the electrode on each test.

**Multimodality Tests**: The multimodality logic we apply (in order of stringency) is as follows: (1) Is any multimodal model or linearly-integrated vision-language model significantly more predictive than all other unimodal models in *either* of our dataset alignments (word onset, scene cuts)? (2) Is the SLIP-Combo vision transformer significantly more predictive than the SLIP-SimCLR vision transformer in *either* of our dataset alignments? (3) Is any multimodal model or linearly-integrated vision-language model significantly more predictive than all other unimodal models in BOTH of our dataset alignments? (4) Is the SLIP-Combo vision transformer more predictive than SLIP-SimCLR vision transformer in BOTH of our alignments? (5) For electrodes that pass test 3, is a multimodal model more predictive than both linearly-integrated vision-language models i.e. MultiConcat and MultiLin? (A more detailed description and reasoning behind these choices is given in Appendix D). For these tests, we use both the "default winner analysis" (i.e. an electrode passing automatically if the only model left after filtering is either multimodal or SLIP-Combo more specifically), and the bootstrapped model comparison test. Tests 2 and 4 control for architecture and dataset, which ensures that models cannot be outperformed due to architecture, hyper-parameters, or the training dataset. For all electrodes that pass our multimodality test, we use our model comparisons to identify the best multimodal architecture for explaining activity in the electrode.

## 4 RESULTS

While there is no single meaningful measure of overall modeling performance, since we expect significant variance in performance as a function of *multiple* controlled and uncontrolled sources, there are a few key metrics we can consider to provide an overall gestalt of our model-to-brain encoding pipeline and the specific measured effects. Unless otherwise noted, we use the following convention in the reporting of these metrics: arithmetic average over the bootstrapped mean scores [lower 95% confidence interval; upper 95% confidence interval].

As an initial heuristic, we consider the bootstrapped test set score mean, as well as the bootstrapped test mean upper and lower bounds on performance across all N = 28 models (14 architectures, with

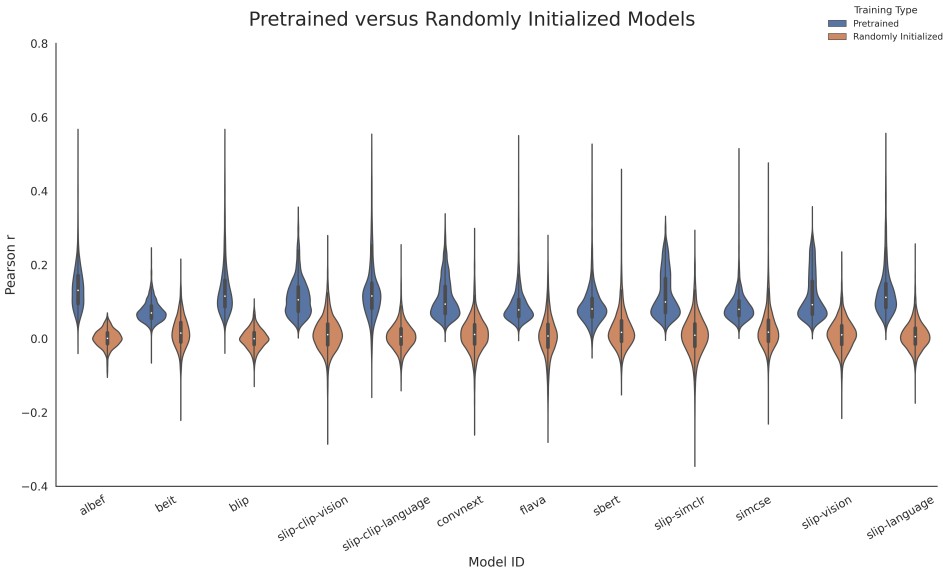

Figure 2: A comparison between pretrained and randomly initialized model performance showing the distribution of predictivity across electrodes. This averages significant time bins per electrode, i.e., the lower validation confidence interval must be larger than zero, for both vision and language alignments for our 12 models. Every trained network outperforms its randomly initialized counterpart. Trained networks overall outperform untrained networks. This is true both on average, and for almost every single electrode.

both trained and randomly-initialized weights), N = 2 dataset alignments (word onsets, scene cuts) and all N = 1090 electrodes, after we've selected the max accuracy across time. This constitutes a total of 24 * 2 * 1090 = 39,420 data points. The bootstrapped global average (i.e. an average across the bootstrapped means) across these data points is $r_{\text{Pearson}} = 0.142$ [0.0797, 0.269]. The bootstrapped max across these data points is $r_{\text{Pearson}} = 0.539$ [0.517, 0.561]. And the bootstrapped lower bound (i.e. the bootstrapped minimum) is $r_{\text{Pearson}} = -0.223$ [-0.398, -0.034]. (Negatives here mean model predictions were anticorrelated with ground truth.) This is of course a coarse metric, meant only to give some sense of the encoding performance overall, and to demonstrate its notable range across electrodes.

## 4.1 TRAINED VERSUS RANDOMLY INITIALIZED RESULTS

We first use the comparison methods to analyze the difference between neural predictivity of trained models and neural predictivity of randomly initialized models. After filtering out models and time bins in electrodes where the lower validation confidence interval score is less than zero, this leaves us with 498/1090 unique electrodes. We show the average difference in performance for these electrodes in Figure 2. In 120 of these electrodes, the default model was a trained model after filtering according to the default winners analysis. For the rest of the 278 electrodes, we use a bootstrapping comparison on the remaining electrodes assuming models have at least 10 time bins remaining. We find that trained models beat randomly initialized models on all 278 electrodes according to the bootstrapping comparison. The average difference in scores across the dataset alignments was $r_{\text{Pearson}} = 0.107[0.026, 0.238]$ showing the significant improvement that trained models have on randomly initialized models. These results demonstrate that experience and structured representations are necessary to predict neural activity in our case for any network, regardless of whether the network is a language network, visual network, or multimodal network.

## 4.2 MULTIMODALITY TEST RESULTS

Using our multimodality tests to evaluate the predictive power of multimodal models against unimodal models across the two dataset alignments, the following results were obtained: The first test showed that 213/1090 (19.5%) and 60/1090 (5.50%) electrodes were more predictive using language- and vision-aligned event structures respectively, with average performance differ-

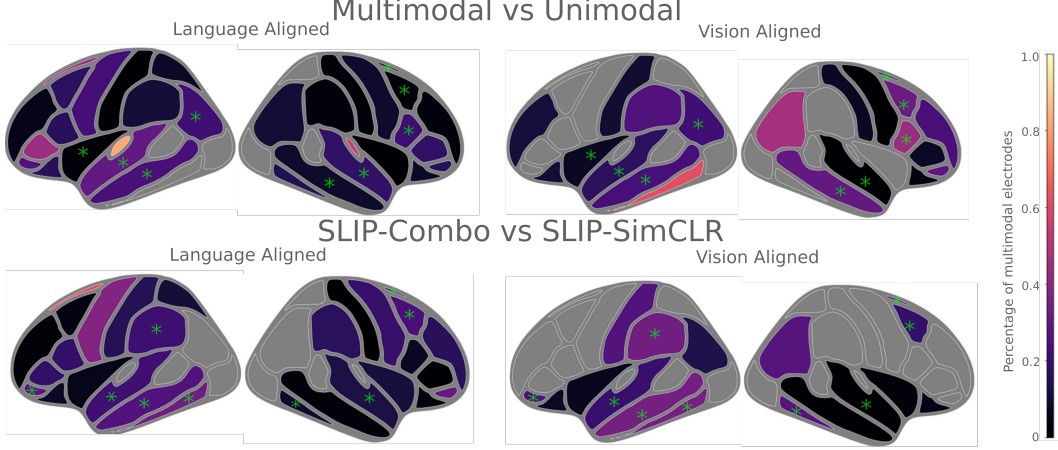

✳ = region contains electrode that is multimodal with both language- and vision-aligned data

Figure 3: Multimodal sites aggregated into regions from the DKT atlas. For each site we compute the percentage of multimodal electrodes using the first test and the (left) language or (right) vision alignment. The top defines a site to be multimodal if the best model that explains that electrode is multimodal as opposed to unimodal. The bottom controls for architecture, parameters, and datasets by comparing SLIP-Combo and SLIP-SimCLR. Gray regions have no multimodal electrodes. Regions which have at least one electrode that is multimodal both with the vision and language aligned stimuli are marked with a green star. We notice that many electrodes occur in the temporoparietal junction with a cluster in the superior temporal cortex, middle temporal cortex, inferior parietal lobe, etc. Other areas we identify include the insula, supramarginal cortex, the superior frontal cortex, and the caudal middle frontal cortex.

ences of $r_{\text{Pearson}} = 0.082[0.011, 0.21]$ and $0.081[0.016, 0.344]$. The second test yielded 218/1090 (20%) and 73/1090 (6.70%) electrodes for language- and vision-aligned structures, respectively, with performance differences of $r_{\text{Pearson}} = 0.046[0.01, 0.140]$ and $0.024[0.01, 0.04]$ between SLIP-SimCLR and SLIP-Combo vision transformers. The third test found 12/1090 (1.1%) electrodes were more predictive in both alignments, with average differences of $r_{\text{Pearson=0.0766}}[0.013, 0.163]$ and $0.0922[0.019, 0.304]$. The fourth test showed 28/1090 (2.57%) electrodes favored the SLIP-Combo over the SLIP-SimCLR in both alignments, with differences of $r_{\text{Pearson}} = 0.0522[0.011, 0.10]$ and $0.026[0.0162, 0.044]$. The final test reiterated the 12/1090 electrodes from the third test, showing a consistent preference for multimodal models over MultiConcat and MultiLin, with performance differences of $r_{\text{Pearson}} = 0.0566[0.025, 0.113]$ and $0.084[0.029, 0.21]$ in the language- and vision-aligned datasets, respectively.

In examining the DKT atlas in Figure 3, it's evident that the largest cluster of multimodal electrodes is around the temporoparietal junction, aligning with previous studies. Key regions include the superior and middle temporal cortex, the inferior parietal lobe, and the supramarginal gyrus, which are close and theoretically linked to vision-language integration. These areas, crucial for tasks like auditory-visual processing, emotion processing, and social cognition, support our findings and previous theories. The multimodal abstractions at this junction might explain their better prediction by multimodal representations. Additionally, electrodes passing tests 3 and 4 in the frontal and prefrontal cortex, specifically in the superior frontal lobe, caudal middle frontal cortex, and pars orbitalis, suggest complex cognitive processing in vision-language integration. This indicates a widespread brain network involved in this integration, corroborating our results and existing literature.

Our multimodality tests demonstrate that multimodal models can greatly out-perform unimodal models at predicting activity in the brain, sometimes by close to $r_{\text{Pearson}} = 0.1$ at some electrodes. This potentially demonstrates that multimodality could be an important factor in improving connections between deep networks and the brain. Furthermore, the areas we identify have commonly been associated with specific forms of vision-language integration identified in prior analyses. These prior analyses were constrained by smaller datasets with strong controls. We reduce these controls and still meaningfully identify the same areas for future study. More specifically, our analyses allow us to study vision-language integration without committing to a specific structural hypothesis. Despite this more general search space, we find overlap with prior work.

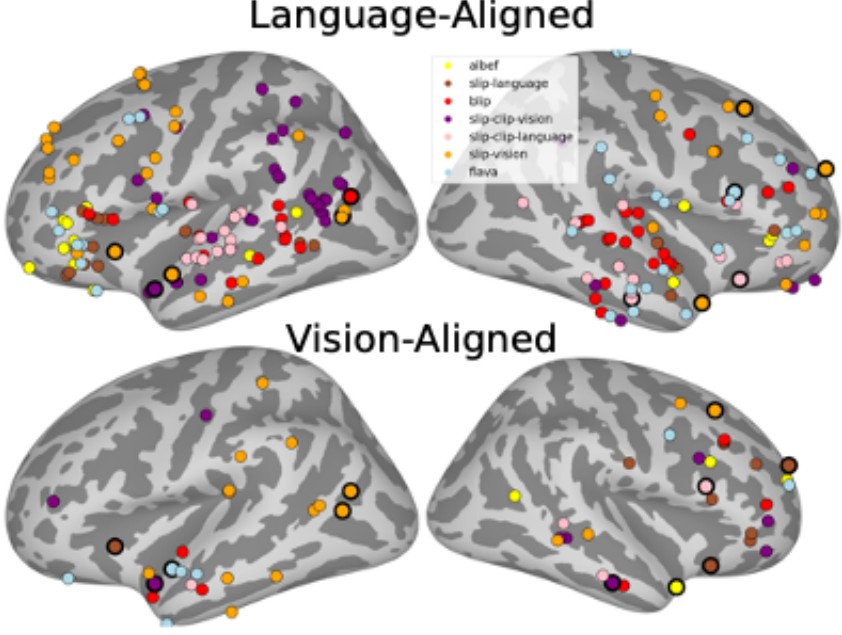

Figure 4: We visualize the individual electrodes that pass our multimodality tests for the language-aligned (top) and vision-aligned datasets (bottom), adding a bold outline to electrodes that pass across both datasets. We color the electrodes by the top-ranked multimodal model that predicts activity in the electrode. We see that models such as SLIP-Combo and SLIP-CLIP often predict activity the best across datasets. We also see that BLIP and Flava are the best architecturally multimodal models.

## 4.3 MODEL TASK PERFORMANCE

| Model | Next-Word Perplexity (↓) |
|---|---|
| Average Unimodal | 133.4 |
| Average Multimodal | 210.3 |
| | **Scene Class Accuracy (↑)** |
| Average Unimodal | 74.2 |
| Average Multimodal | 54.3 |

Table 1: We report average unimodal task performance for unimodal models and multimodal models. We show next-word prediction perplexity and scene-cut class accuracy for one movie. Our findings demonstrate that unimodal models have better unimodal representations than multimodal models as reflected by better performance.

While our study aims to explore vision-language integration, we must consider other explanations, such as whether our multimodal networks outperform unimodal networks in language or visual reasoning. This could imply that our findings are more about unimodal processing than vision-language integration. To address this, we evaluated our multimodal networks' performance on unimodal tasks. For language tasks, we assessed next-word prediction in both multimodal and unimodal language networks, using perplexity as a metric. For vision tasks, we tested scene classification abilities using the Places365-labeled dataset. Our results, detailed in Table 1, show that multimodal networks perform worse on unimodal tasks compared to unimodal networks, reducing the likelihood that our findings are merely improvements in unimodal representation by multimodal networks.

## 4.4 WHICH MULTIMODAL ARCHITECTURE IS MOST "BRAIN-LIKE"?

In Figure 4, we use our model ranking techniques and show the most predictive multimodal model for each of the electrodes that pass our tests of multimodality (specifically, tests 1, 3, and 5). We see that consistently, trained multimodal models such as SLIP-Combo and SLIP-CLIP are the most predictive multimodal model in comparison to architecturally multimodal models such as ALBEF, BLIP, or Flava. We also find that the SLIP-Combo and SLIP-CLIP language encoders are the most predictive networks in the vision-aligned data indicating that visual features are captured in the representations of these networks in some manner. While the vision encoder has been commonly

benchmarked, this could indicate that CLIP-style pretraining is meaningful in building grounded language representations that are similar to the brain.

We do see that more architecturally multimodal models predict the language-aligned dataset and generally find that BLIP is the most predictive architecturally multimodal model. There are many possible reasons why models like SLIP-Combo or SLIP-CLIP out-predict these architecturally multimodal models such as dataset considerations or the need for better cross-attention design. These results could indicate that the cross-attention mechanism used in models such as BLIP is better at integrating vision and language in a manner similar to the brain. This allows for more future work in this direction.

In general, our findings show that network parameter size does not correlate with predictivity. SLIP-Combo and SLIP-CLIP have fewer parameters than our architecturally multimodal models and even our unimodal models. This indicates a special feature in CLIP-style training that can be studied more carefully in future work.

## 5 CONCLUSION

The methodology introduced here provides a fine-grained analysis that overcomes a first hurdle: it distinguishes randomly initialized and trained language networks in every modality individually and then across modalities. Having overcome this hurdle, we can now identify areas which are better explained by multimodal networks compared to unimodal networks and linearly-integrated language-vision networks. The most-fine grained result we provide compares SLIP-Combo vs SLIP-SimCLR, a multimodal and unimodal network controlling for architecture, dataset, and parameter count. To enable future research, we release a toolbox for multimodal data analysis along with, upon request, the raw neural recordings under a permissive open license such as Creative Commons.

We identified a cluster of sites which connect vision and language[1] . This appears to be a network which spans the temporoparietal junction, connecting the superior temporal cortex, middle temporal cortex, inferior parietal lobe, and supramarginal gyrus, to areas in the frontal lobe, containing the pars orbitalis, superior frontal cortex, and the caudal middle frontal cortex. These areas align with prior studies and analyses on particular aspects of vision-language integration in the brain. Given a lack of ground truth for general vision-language integration, we hope this serves as a starting point for more study.

While our data has high fine-grained temporal resolution, and our method is sensitive to the time course of the signal, our final analysis aggregates across time bins. We have not investigated how multimodal integration occurs as a function of time. This could be used to derive a time course of integration across the brain, to establish a network structure, and to find other distinctions between areas.

Our method is agnostic to the modalities used, or to the number of modalities. Neural networks exist which integrate not just language and vision, but also audio and motor control. These could also be used with our method (our data even explicitly enables future modeling with audio). The distinction and interaction between audio processing and language processing could be revelatory about the structure of regions which transition from one to the other, like the superior temporal gyrus.

While prior work has investigated which network architectures, motifs, and hyper-parameters are most brain-like when explaining visual activity in the brain, and to a lesser degree language, we do not know the equivalent for multimodal networks. There are a wide range of architectures available, from CNNs, to multi-stream Transformers which perform late integration. Our approach could be used to determine which of these approaches is most brain-like, helping to guide future research. One could even imagine this approach enabling a variant of Brain-Score dedicated to multimodal processing in order to systematically investigate this over time as a community.

---

[1]This work was carried out under the supervision of an Institutional Review Board (IRB). 28 modern GPUs on 7 machines were used for four weeks, evenly distributed across experiments.

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

# A  NEURAL DATA DETAILS

## A.1  DATA COLLECTION OVERVIEW

## A.2  EVENT STRUCTURES

We parse our neural activity into individual language and single movie-frame combinations (which we call interchangeably *event structures* or *text-image pairs*) by discretizing the movie stimulus,

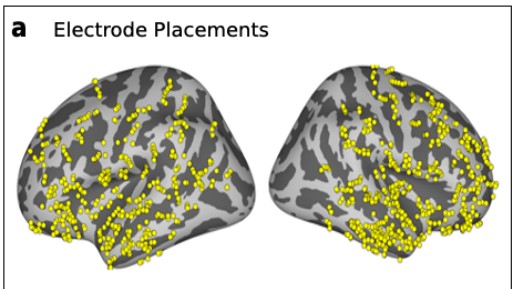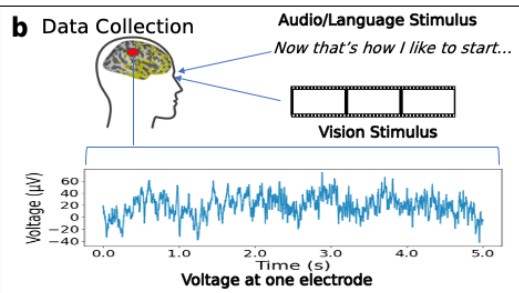

Figure 5: (a) The electrode placements over all subjects. Each yellow dot denotes an electrode collecting invasive field potential recordings for further analysis in our experiments. (b) An overview of our data collection procedure. Subjects are presented feature length films while neural data is collected from these electrodes in the brain.

| Subj. | Age (yrs.) | # Electrodes | Movie | Recording time (hrs) |
|---|---|---|---|---|
| 1 | 19 | 154 | Fantastic Mr. Fox | 1.83 |
| 2 | 12 | 162 | Venom | 2.42 |
| 3 | 18 | 134 | Cars 2 | 1.92 |
| 4 | 6 | 156 | Fantastic Mr. Fox | 1.5 |
| 5 | 16 | 162 | Sesame Street Episode | 1.28 |
| 6 | 4.5 | 106 | Ant Man | 2.28 |
| 7 | 12 | 216 | Cars 2 | 1.58 |

Table 2: **Subject statistics** Age (second columns), number of electrodes (third column), movie shown (fourth column) and recording time (fifth column) per subject. Electrode placements are done for clinical purposes and the distribution of electrode locations differ from subject to subject. The average amount of recording data per subject is 1.83 (hrs).

allowing us to feed inputs to our deep neural network models (which are not trained on movie data). We define event structures by the guiding feature used to select a particular text-image pair in the movie for analysis. So as not to unfairly prioritize one modality over the other or impose a hypothesis over vision-language integration, we design two different kinds of event structures: The first kind of event structure consists of word onset times, a language-aligned event. Word onsets have been used in prior work (Goldstein et al., 2021) and are commonly associated with language processing. For each word onset, we take the prior sentence context of the given word to add contextual information for the language models. We also take the closest frame after the word onset as the associated image input. The second kind of event structure consists of visual scene cuts (i.e. camera cuts). We extract the frames associated with a scene cut as proxy for visual processing given a shift in the pixel distribution between frames. We then take the closest sentence that occurred after the scene cut. (Note that by language-alignment or vision-alignment, here, we mean the anchoring of points in neural time-series to points in the movie).

We use these two kinds of event structures to create two datasets. Our language-aligned dataset consists of [context of a given word, closest frame pairs] with the associated neural activity as processed in Section 3. Our vision-aligned dataset consists of [scene cut frames, closest sentence to a scene cut frame] with similar processing on the neural activity. We analyze all results over the datasets individually and then compare results across the datasets to identify electrodes for multimodal integration.

We note that our two datasets cover many possible hypotheses of vision-language integration. The language-aligned dataset likely covers short-term integration as each corresponding context segment

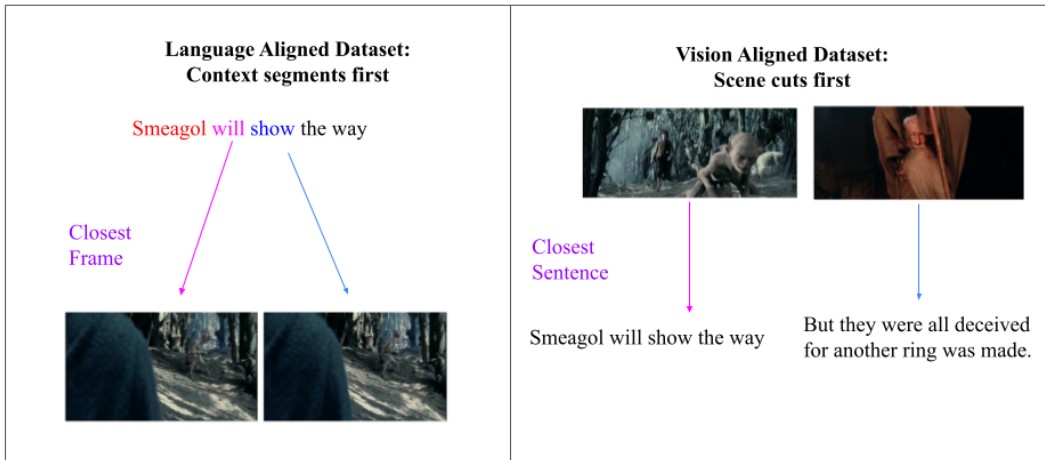

Figure 6: We construct two datasets for understanding vision-language integration in the brain. (Left) The language-aligned dataset consists of choosing context segments and the closest frame. Context segments choose each word and the corresponding sentence context. (Right) The vision-aligned dataset consists of choosing a scene cut and the closest sentence that occurs after a scene cut.

has a nearby frame. However, our vision-aligned dataset likely covers long-term integration since there is separation between the scene cut and corresponding sentence. This design makes our experiments more difficult by considering many forms of possible visio-linguistic reasoning.

### A.3 NOTES ON STIMULUS INDEPENDENCE (AUTOREGRESSION)

Converting neural activity measured in response to naturalistic movie-viewing to a dataset of nominally IID event-structures presents a particular challenge often explicitly avoided in experimental designs that leverage otherwise unrelated natural images or language prompts: that is, nonindependence in the form of autoregression. Movies (driven as they are by common visuolinguistic themes) contain inherently autoregressive structure that can lead to overfitting in parametrized predictive models designed to predict neural response patterns evoked by that structure. The parsing of our final event-structures into training, testing and validation splits was designed explicitly to assess for such overfitting. When creating the train-validation-test splits, we assign contiguous chunks of the movie to each split. In practice, and especially for movies with more linear narrative structure, we assumed this continuous splitting could provide at least a weak form of independence between sampled event-structures. While this by no means fully accounts for the non-independence of the stimulus set writ large, our results across the training, validation, and test splits suggest that it does help to minimize potential overfitting. In future work, we hope to revisit our event-structure delineation and sampling, potentially leveraging movie-trained models like Salesforce's ALPRO (Li et al., 2022a) to select stimuli that are more distinct not just at the level of pixels or words, but in latent feature space.

## B CANDIDATE DEEP NEURAL NETWORK MODELS

We present a full set of networks in Table 3. Because they control for dataset and architecture (varying only the learning objective), comparisons amongst the variants of the SLIP models are our most empirically rigorous test of multimodality.

However, given that the SLIP models contain only one kind of multimodal - unimodal contrast (SLIP-SimCLR versus SLIP-Combo's visual encoder), we added a number of uncontrolled model contrasts to assess the predictive power of unimodal and multimodal representations more generally. These models include ALBEF (Li et al., 2021) (a two channel multimodal encoder that uses a vision transformer and language transformer trained with a contrastive loss followed by a multimodal transformer); BLIP (Li et al., 2022b) (a two channel multimodal enoder similar to ALBEF but trained with an image-text matching loss and momentum model); Flava (Singh et al., 2022) (a two channel multimodal encoder with a multimodal encoder that builds fused embeddings and trained

| Model | Modality | Architecture | Parameters |
|---|---|---|---|
| ALBEF | Architecturally Multimodal | Transformer | 209.8M |
| BLIP | Architecturally Multimodal | Transformer | 223.5M |
| Flava | Architecturally Multimodal | Transformer | 241.4M |
| SLIP-Combo Vision | Trained Multimodal | Transformer | 86M |
| SLIP-Combo Language | Trained Multimodal | Transformer | 63.6M |
| SLIP-CLIP Vision | Trained Multimodal | Transformer | 86M |
| SLIP-CLIP Language | Trained Multimodal | Transformer | 63.6M |
| SBERT | Language | Transformer | 109.5M |
| SimCSE | Language | Transformer | 109.5M |
| SLIP-SimCLR | Vision | Transformer | 86M |
| BEIT | Vision | Transformer | 86.3M |
| ConvNeXt | Vision | Convolutional | 109.9M |

Table 3: A catalogue of the networks we include in this experiment. We compare architecturally multimodal networks, trained multimodal networks, unimodal language and unimodal vision networks. We mostly study transformers but also include ConvNeXt, a CNN model. We tabulate the number of parameters in the model.

reconstructively); SBERT (Reimers & Gurevych, 2019a) (a unimodal masked language transformer for sentence embeddings); BEIT (Bao et al., 2021) (a unimodal vision transformer trained via masked image reconstruction); SimCSE (Gao et al., 2021) (a unimodal language transformer trained via contrastive learning); ConvNeXt (Liu et al., 2022b) (a unimodal vision convolution network built by modifying the ResNet architecture). These models provide a broader sample of multimodal and unimodal networks, while still maintaining some core similarities with the SLIP models (transformer backbones or contrastive learning.)

We also introduce two *linearly integrated vision-language models*, *MultiConcat*, and *MultiLin*. Multi-Concat consists of concatenating representations from SimCSE and SLIP-SimCLR, and MultiLin extends MultiConcat by introducing a trained linear projection to project the concatenate representation to a dense vision-language vector trained using the NLVR-2 dataset (Suhr et al., 2018). By introducing these models, we aim to distinguish between areas that are simply responding to the presence of vision and language features and areas that are integrating vision and language in a rich, non-linear fashion using comparisons we describe more in detail.

We assess both trained and randomly-initialized versions of these models first and foremost because, in most cases, the multimodality of these models is a function ONLY of their learning objective: This means, for example, that models like the SLIP models – which consist of architecturally encapsulated vision and language encoders – cannot, in the absence of training, be considered multimodal. Models like ALBEF, BLIP, or Flava, on the other hand, may be considered multimodal even in the absence of training due to architectural inductive biases such as cross-modal attention-heads that integrate linguistic and visual inputs from the outset of processing.

## C NEURAL REGRESSIONS

In this section, we detail our neural regression pipeline, which proceeds in 4 phases: feature extraction, dimensionality reduction (via sparse random projection), cross-validated ridge regression, and scoring.

### C.1 FEATURE EXTRACTION

This follows from approaches taken in Conwell et al. (2021). We consider feature extraction to mean the extraction of a separate feature vector at *every layer* in a network – in other words, each distinct tensor operation module that progressively transforms model inputs into outputs. This means, for example, that we consider not only the outputs of each transformer attention head, but also of the individual key, query, value computations that produce them. If the layer has more than 1 dimension, then we flatten the tensor such that each layer represents any given input as a 1-dimensional feature vector. (Note: This flattening makes no assumptions about the separation of a given feature space into

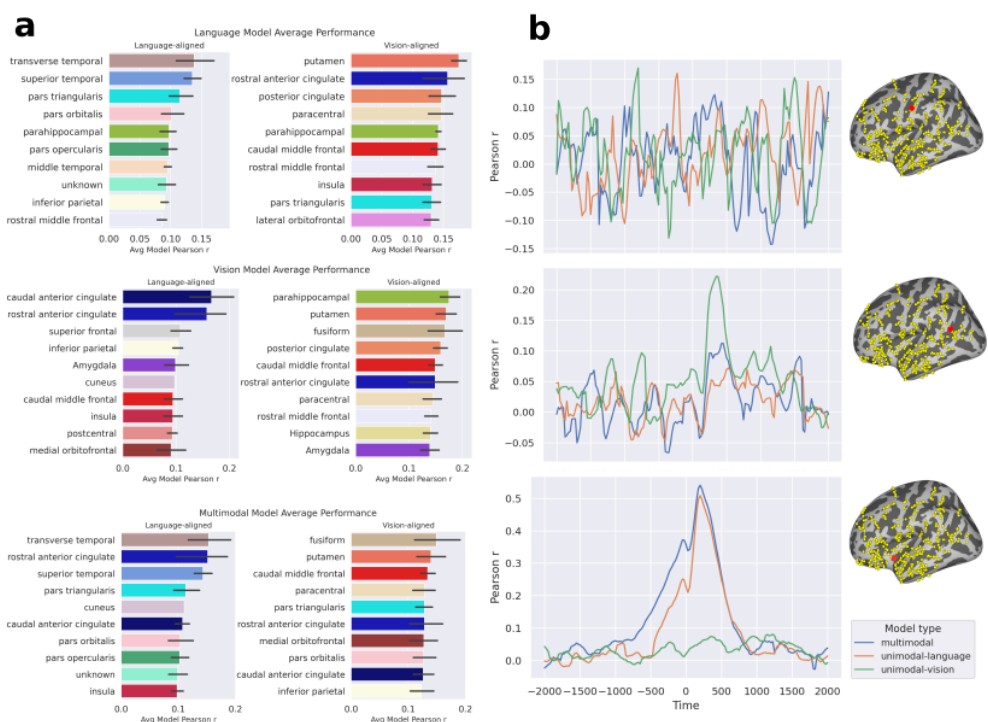

Figure 7: (a) The top ten Desikan-Killiany-Tourville (DKT) regions, ranked based on average predictivity, Pearson r, across electrodes in that region. Error bars represent the standard error associated with averaging the Pearson r over the electrodes and time bins. All results shown here are from pretrained variants of the model. (top) A language model, SBERT. (mid) A vision model, SLIP-SimCLR. (bottom) A multimodal model, BLIP. (b) The predictivity of these same three models for three typical electrodes. (top) An arbitrary electrode which is not responsive to language or vision. (mid) An electrode which is responsive to vision, but not language, and is not better explained by multimodal integration. (bottom) An electrode which is responsive to language and that is better explained by a multimodal network. Confidence intervals are not shown for clarity.

spatial and semantic components, and allows the subsequent regression to reweight all contributing components as relevant). The output tensor thus constitutes a dataset of $n$ inputs (either images, sentences, or image-sentence pairs) as an array $\boldsymbol{F} \in \mathbb{R}^{n \times D}$ where $D$ is the dimensions of the feature vector.

## C.2 Sparse Random Projection

For certain flattened feature vectors, the dimensionality $D$ is very large, and as such performing ridge regression on $\boldsymbol{F}$ is prohibitively expensive, with at best linear complexity with $D$, specifically $\mathcal{O}(n^2 D)$ (Hastie & Tibshirani, 2004). We use the Johnson-Lindenstrauss lemma (Johnson, 1984; Dasgupta & Gupta, 2003) to project $\boldsymbol{F}$ to a low dimensional representation $P \in \mathbb{R}^{n \times p}$ that preserves pairwise distances in $\boldsymbol{F}$ with errors bounded by a factor $\epsilon$. If $u$ and $v$ are any two feature vectors from $\boldsymbol{F}$, and $u_p$ and $v_p$ are the low-dimensional projected vectors, then

$$(1 - \epsilon)||u - v||^2 < ||u_p - v_p||^2 < (1 + \epsilon)||u - v||^2 \tag{1}$$

Equation 1 holds provided that $p \geq \frac{4 \ln(n)}{\epsilon^2/2 - \epsilon^3/3}$ (Achlioptas, 2001). To find the mapping from $\boldsymbol{F}$ to $\boldsymbol{P}$, we used *sparse random projections* (SRPs) following Li et al. (2006). The authors show a $\boldsymbol{P}$ satisfying Equation 1 can be found by $\boldsymbol{P} = \boldsymbol{F}\boldsymbol{R}$ where $\boldsymbol{R}$ is a sparse $n \times P$ matrix with i.i.d. elements shown below:

$$r_{ij} = \begin{cases} \sqrt{\frac{\sqrt{D}}{p}} & \text{with prob. } \frac{1}{2\sqrt{D}} \\ 0 & \text{with prob. } 1 - \frac{1}{\sqrt{D}} \\ -\sqrt{\frac{\sqrt{D}}{p}} & \text{with prob. } \frac{1}{2\sqrt{D}} \end{cases} \tag{2}$$

If $\boldsymbol{F}$ has dimensionality $D$ that is less than the dimensionality of the Johnson-Lindenstrauss lemma, then no projection is applied. In this case, $\boldsymbol{P} = \boldsymbol{F}$.

## C.3 $k$-fold Ridge Regression

To determine how well vision and language networks predict activity in the brain, we ran regressions from representations extracted from a specific layer of either a multimodal or unimodal network to predict the average activity of the SEEG signals over a window of time for all electrodes of our 7 subjects. We detail the steps we took to run regressions per subject below.

We use ridge regression to predict the average activity, $\boldsymbol{y}$, at a given electrode and time point as constructed in Section 3, from their associated DNN features $\boldsymbol{P}$. Given the sequential nature of our data, we used a 5-fold cross-validation procedure. For each fold, we split our dataset of representations into a contiguous training set(80%), $\boldsymbol{P}_{\text{train}}$ and $\boldsymbol{y}_{\text{train}}$, a contiguous validation set (10%), $\boldsymbol{P}_{\text{valid}}$ and $\boldsymbol{y}_{\text{valid}}$, and contiguous testing set (10%), $\boldsymbol{P}_{\text{test}}$ and $\boldsymbol{y}_{\text{test}}$. Each split takes a contiguous chunk of event structures in order of their occurrence in the movie, and each fold changes the starting point of the training, validation, and testing set such that different contiguous chunks are assigned to a different set. We standardize the columns of $\boldsymbol{P}_{\text{train}}$ and $\boldsymbol{P}_{\text{valid}}$ to have mean 0 and a standard deviation of 1 and fit this standardization on $\boldsymbol{P}_{\text{test}}$. We fit the coefficients $\hat{\beta}_i$ of a regression model on the train dataset such that $\boldsymbol{y}_{\text{train}} = \boldsymbol{P}_{\text{train}}\hat{\beta}_i + \epsilon$ with minimal error $||\epsilon||$. Ridge regression penalizes large $||\hat{\beta}||$ proportional to a hyperparameter $\lambda$, which is useful in preventing overfitting when regressors are high-dimensional and highly correlated. Each $\hat{\beta}$ is calculated by the fixed ridge regression solution:

$$\hat{\beta} = ((\boldsymbol{P}_{\text{train}})^T \boldsymbol{P}_{\text{train}} + \lambda \boldsymbol{I}_d)^{-1} (\boldsymbol{P}_{\text{train}})^T \boldsymbol{y}_{\text{train}} \tag{3}$$

The coefficients $\hat{\beta}$ are then used to predict the held out data where:

$$\begin{aligned} \hat{\boldsymbol{y}_{\text{valid}}} &= \boldsymbol{P}_{\text{valid}}\beta \\ \hat{\boldsymbol{y}_{\text{test}}} &= \boldsymbol{P}_{\text{test}}\beta \end{aligned} \tag{4}$$

We use the *KFold* function from Pedregosa et al. (2011) and implemented ridge regression in Pytorch (Paszke et al., 2019). In this analysis we run the 5-fold regression per $\lambda$ value, where $\lambda$ was varied using a logarithmic grid search over $10^{-1}$ to $10^6$. On each fold, we calculated a *score* for the

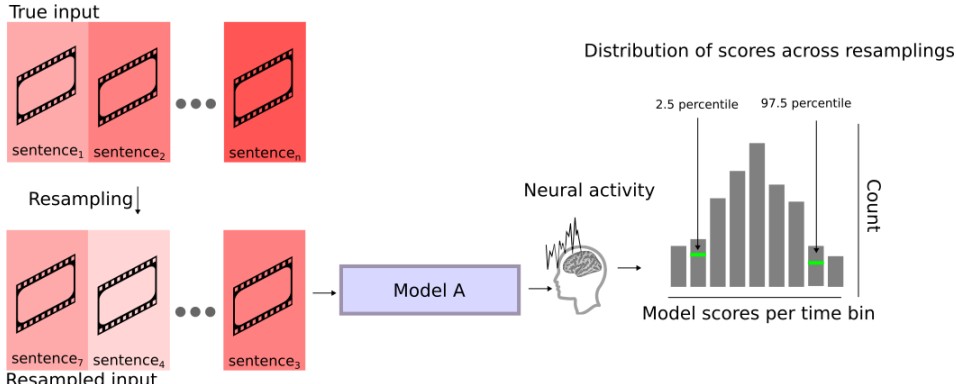

Figure 8: We bootstrap our event structures and corresponding neural activity to derive 95% confidence intervals per time bin per electrode on our training, validation, and testing set in our analysis.

prediction $\hat{\boldsymbol{y}_{\text{valid}}}$ and $\hat{\boldsymbol{y}_{\text{test}}}$ by computing the Pearson correlation coefficient. This score is averaged over the $5$ folds to get final validation and test set scores. We choose the best $\lambda$ value using the cross-validated scores and take the associated test scores with the $\lambda$ value. We run this regression for all electrodes and time points simultaneously.

To analyze network performance over all layers, we select the best performing layer using the validation set. Specifically, per electrode, we average the validation correlation scores over time and take the layer with the max average score. We then take the associated test set correlation score as the overall score per model.

We provide initial results in Figure 7. We first show the top ten Desikan-Killiany-Tourville (DKT) regions based on the average score over one model from the three model types: a unimodal language model, a unimodal vision model, and a multimodal model. We notice several re-occurring regions across the three model types including the superior temporal cortex, the rostral anterior cingulate, the transverse temporal cortex, the fusiform and the superior frontal cortex. These are regions associated with the temporoparietal junction, indicating a potential network of vision-and-language processing with connections to the frontal lobe. Many of these regions are associated with high level emotion processing and spatial processing. Furthermore, we show the results of the three model types on three particular electrodes. The first electrode is in the precentral gyrus and gives the average performance of the models with peak performance of $r_{\text{Pearson}} = 0.15$. The second electrode is in the inferior parietal lobe and gives an example of an electrode with strong performance from vision models. Finally, the last electrode is in the superior temporal lobe and gives an example of an electrode processing language and potentially integrating multimodal features. The analysis will use these scores over time per electrode and compare the score distributions to quantify statistically whether a multimodal model is performing significantly better than a unimodal model.

## D   STATISTICAL TESTING DETAILS

### D.1   BOOTSTRAPPING OVER EVENT STRUCTURES

As shown in Figure 8, we introduce a bootstrapping procedure over our event structures (image-text pairs) which allows us to derive confidence intervals on our regression per time bin $t$ and electrode $e$ in our analysis.

In our bootstrapping procedure, we first resample the image-text pairs and corresponding neural activity with replacement per movie and subject 1000 times. We use the same 1000 resampled image-text pairs and corresponding neural activity across all models (both trained and randomly initialized) to allow for model comparison in later analysis. We denote this resampling with replacement as

follows:

$$\{(\boldsymbol{P}^{(1)}), \boldsymbol{y}^{(1)}), \cdots (\boldsymbol{P}^{(999)}), \boldsymbol{y}^{(999)}), (\boldsymbol{P}^{(1000)}), \boldsymbol{y}^{(1000)})\} \sim \{\boldsymbol{P}, \boldsymbol{y}\} \qquad (5)$$

Most importantly, for each $(\boldsymbol{P}^{(i)}, \boldsymbol{y}^{(i)})$, we sort the resampled indices by their occurrence in the movie time to maintain the autoregressive structure of data. We note that the scores are skewed upwards when we do not sort.

We then rerun the regression for each $(\boldsymbol{P}^{(i)}, \boldsymbol{y}^{(i)})$ to derive the 95% confidence intervals by taking the 2.5th percentile and 97.5th percentile on our score distribution per $t$ per $e$ on the training, validation, and testing set. Upon inspection of bootstrapping scores on a sample of time bins and electrodes, we note that the scores are normally distributed.

We use this information as a filter, identifying time-bins and electrodes that do not have meaningful response to our event structures. We interpret the lower confidence interval of our validation set bootstrapping scores as the lower bound on our score parameter [2]. For a time bin where this lower bound is below 0, we can say that no meaningful mapping to the brain has been learned.

### D.2   MODEL COMPARISONS

Each of our multimodality tests in the main analysis is predicated on a model comparison procedure. After our bootstrapping procedure, we filter the time bins in electrodes for models where the lower 95% confidence interval of the validation score overlapped with 0 (see above). Furthermore, we remove models from further analysis on a particular electrode if all time bins of the model has a validation confidence interval that overlaps with 0 for all time bins.

Per electrode $e$, we identify models that have at least 10 time bins where the lower validation confidence interval is greater than zero. If there are no other models with 10 such time bins, then only this model remains for further analysis and we refer to such models as "default winners", meaning the model has the default best performance on the electrode and has statistically significant performance on an electrode.

For electrodes that did not have a "default winner", we design a model ranking based on a comparison of the mean bootstrapped validation score over time. We define a confidence interval on the bootstrapped validation scores over time per model on each electrode and sort the mean bootstrapped validation scores over time to obtain a model ranking.

Using the first and second highest ranking models, we use a second-order bootstrapping procedure across time bins in $e$ rather than event structures. We only compare the top two models as these are the best predictors of the activity in $e$ and we assume that if the first ranking model is significantly better than then second ranking model, then this significant difference will hold with all other models. We calculate the difference in the average score across the 10 or more resampled time bins between the 2 candidate models in a given comparison. This procedure gives us a p-value per model comparison on each electrode. We repeat this procedure for all electrodes for each of the model comparison tests we describe below. We then use FDR (Benjamini-Hochberg) (Thissen et al., 2002) multiple comparisons corrections to adjust the $p$-value associated with each test on each electrode.

Each of the 5 tests we conduct are suggestive of multimodality, but each successive test provides additional evidence. For tests 1 and 2, we only consider results per dataset alignment. Test 1 considers all comparisons but test 2 controls for architecture and training details with SLIP-Combo and SLIP-SimCLR. Since results in tests 1 and 2 could be explained by unimodal task performance, we introduce tests 3 and 4 where we identify sites that are multimodal in both dataset alignments. Such sites must be predictive by models that perform vision tasks and language tasks. In particular, we can note that test 3 and 4 are identifying sites where multimodal models outperform all unimodal language and unimodal vision models on all possible vision and language tasks associated with a particular neural site. We introduce test 5 as a test for finding the comparing types of multimodal integration, i.e. comparing rich integration styles versus simple linear integration styles where vision and language features are present but not integrated. After multiple comparison corrections, we tabulate the total number of electrodes that significantly pass each test as a proportion of the total number of assayed electrodes (N=1090). After aligning the location of the various electrodes to the

---

[2]We emphasize that this is done on our validation set to allow for unfiltered comparison on our test set scores.

regions provided by the Desikan-Killiany-Tourville atlas (Klein & Tourville, 2012), we can further subdivide this proportion by the number of electrodes located in each region.

# E    MODEL TASK PERFORMANCE

We give an overview of model task performance by reporting the accuracy of our 12 candidate models on several language-, vision- and multimodal-related tasks. We only include results on tasks where the model was evaluated under the same setting e.g. zero-shot task performance. Our multimodal model performance can be seen in Table 4. The SLIP model performance and unimodal vision model performance, which is only evaluated on ImageNet can be seen in Table 5. Our unimodal language model performance is reported in Table 6. We do not report the performance the SLIP model language encoders since they have not been evaluated to our knowledge.

| Model | VQA-v2 | NLVR$^2$ | Flickr30K (1K Test Set) Text Retrieval R@5 | Flickr30K Image Retrieval R@5 |
|---|---|---|---|---|
| BLIP | 77.5 | 82.3 | 99.7 | 96.7 |
| Flava | 72.5 | 78.9 | 94.0 | 89.38 |
| ALBEF | 75.8 | 83.1 | 99.5 | 96.3 |

Table 4: Multimodal model zero-shot performance on 4 multimodal tasks. These include the VQA-v2 dataset (Goyal et al., 2017), the NLVR$^2$, dataset (Suhr et al., 2018), and the Flickr30K dataset (Plummer et al., 2015).

| Model | ImageNet Linear | ImageNet Finetuned |
|---|---|---|
| SLIP-Combo | 72.1 | 82.6 |
| SLIP-CLIP | 66.5 | 80.5 |
| SLIP-SimCLR | 64.0 | 82.5 |
| ConvNeXt | 83.8 | 86.8 |
| BEIT | 76.5 | 86.3 |

Table 5: ImageNet-1K (Russakovsky et al., 2015) top-1 results across the SLIP models and unimodal vision models. The left column shows performance using linear classification and the right column shows performance from finetuning.

| Model | STS-16 | STS-B | SICK-R |
|---|---|---|---|
| SBERT | 74.3 | 77.0 | 72.9 |
| SimCSE | 80.8 | 84.2 | 80.4 |

Table 6: Unimodal language model performance on the Semantic Task Similarity datasets (Pontiki et al., 2016) and SICK dataset (Marelli et al., 2014)

Furthermore, we report statistics on the stimulus dataset, the Brain Treebank. We train a next-word prediction task using all multimodal and language model and report the test-set perplexity in Table 7. We also measure the scene classification performance over the scene-cuts dataset with our multimodal and vision models and report this in Table 8.

# F    MEDIAL REGION ANALYSIS

We repeat our analysis in Section 4.2 but visualize medial regions in Figure 9. We identify two main medial areas, the isthmus cingulate (Liu et al., 2022a) and the superior frontal cortex (Willems et al., 2009). This aligns with prior findings studying medial areas, identifying integration of vision, language, and action. Future work can include deeper study of medial areas but we do not carry this in our paper because we do not have many electrodes in medial regions, making results noisy.

| Model | Test Set Perplexity (One Movie) |
|---|---|
| BLIP | 168.7 |
| ALBEF | 223.8 |
| Flava | 202.2 |
| SLIP-Combo Language Encoder | 197.6 |
| SLIP-CLIP Language Encoder | 259.3 |
| SBERT | 121.3 |
| SimCSE | 145.4 |

Table 7: Test set perplexity measured on the dialogue of a single movie across all models that have language inputs. We see that unimodal language encoders perform as well or better than multimodal encoders, which shows that our sites of multimodal integration are not associated with better unimodal processing.

| Model | Scene Classification Accuracy |
|---|---|
| SLIP-Combo Vision Encoder | 73.9 |
| SLIP-CLIP Vision Encoder | 71.2 |
| SLIP-SimCLR Vision Encoder | 71.0 |
| BEIT | 75.4 |
| ConvNeXt | 76.2 |
| ALBEF | 60.1 |
| Flava | 58.4 |
| BLIP | 62.4 |

Table 8: Scene-Cut classification accuracy for one movie. Scene-cuts are labeled with the Places2 dataset labels and a linear classifier is trained to take image representations and decode the scene-cut label. We see that our multimodal models perform worse than unimodal vision models.

## G    MULTIMODAL ELECTRODE VISUALIZATION

## H    SLIP-CLIP VS SLIP-SIMCLR

We re-run our architecture and dataset controlled multimodality tests with SLIP-CLIP and SLIP-SimCLR instead of with SLIP-Combo. We show the results in Figure 11.

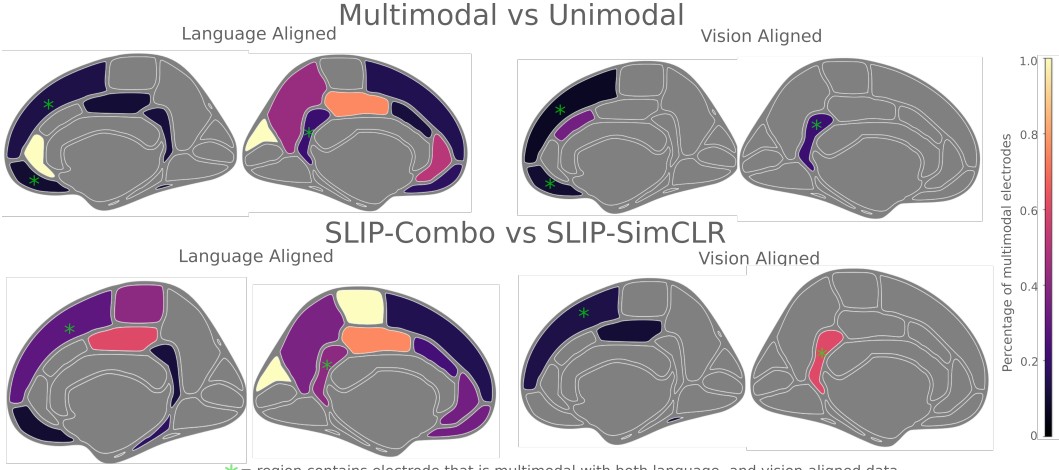

Figure 9: Multimodal sites aggregated into regions from the DKT atlas, visualizing medial regions. For each site we compute the percentage of multimodal electrodes using the first test and the (left) language or (right) vision alignment. The top defines a site to be multimodal if the best model that explains that electrode is multimodal as opposed to unimodal. The bottom controls for architecture, parameters, and datasets by comparing SLIP-Combo and SLIP-SimCLR. Gray regions have no multimodal electrodes. Regions which have at least one electrode that is multimodal both with the vision and language aligned stimuli are marked with a green star. We identify several medial regions including the superior frontal cortex and the isthmus cingulate.

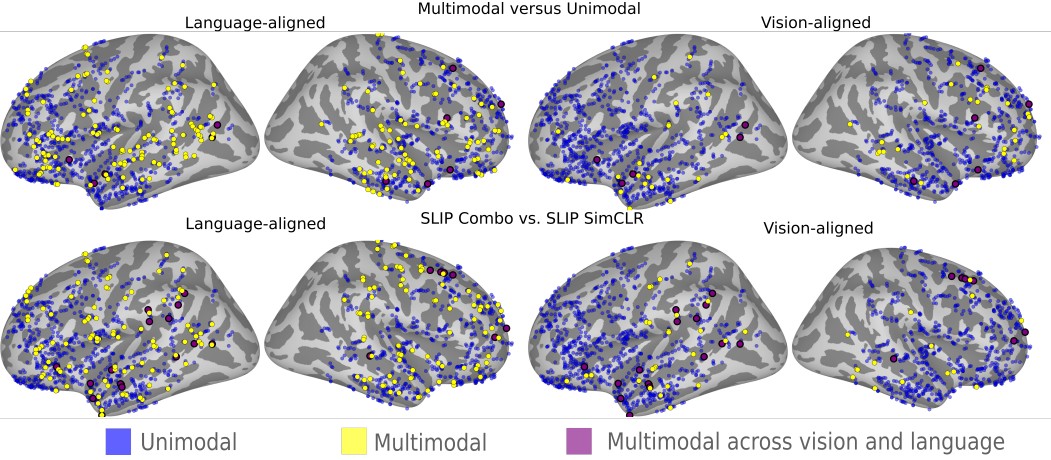

Figure 10: A raw version of Figure 3 and Figure 9 which visualizes the electrode locations instead of aggregating over regions. We multimodal regions in a single modality in yellow and over both dataset modalities in purple.

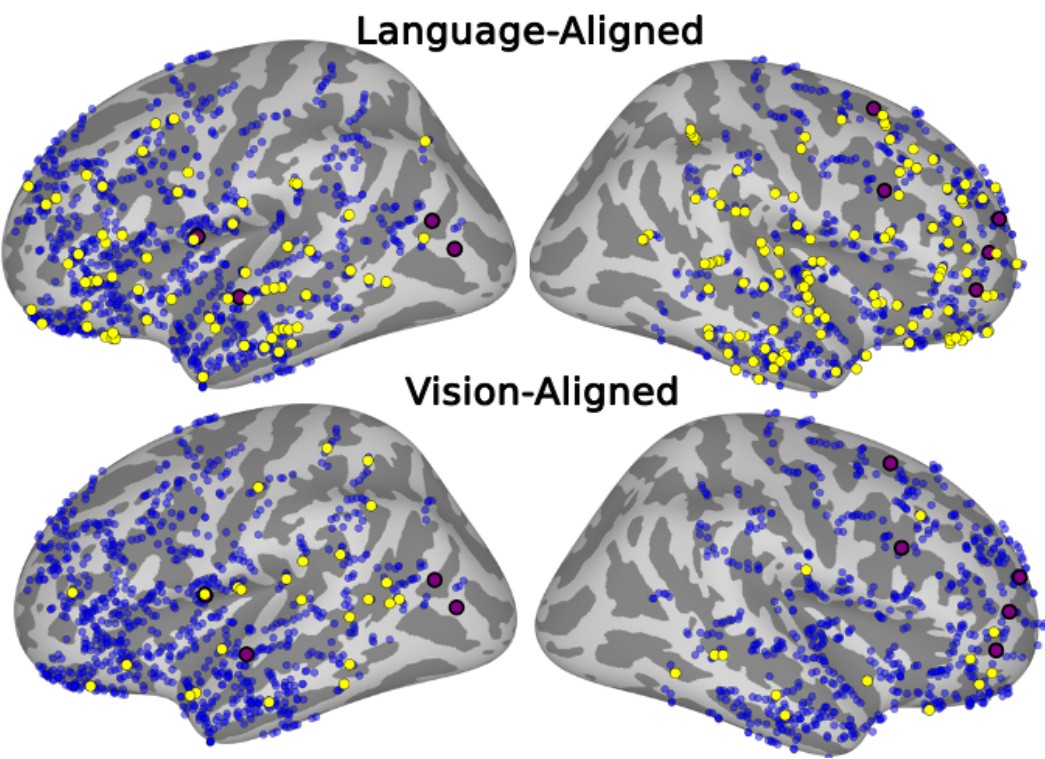

Figure 11: We compare the SLIP-CLIP vision transformer with SLIP-SimCLR instead of using SLIP-Combo. Electrodes that are better predicted by SLIP-CLIP in one dataset alignment are colored yellow. Electrodes that are better predicted by SLIP-CLIP in both dataset alignments are colored purple. We find similar electrodes are better predicted by SLIP-CLIP. A total of 9/1090 electrodes are better predicted by SLIP-CLIP than SLIP-SimCLR in both dataset alignments.

