# OpenReview forum: "Revealing Vision-Language Integration in the Brain with Multimodal Networks"
_ICLR.cc/2024/Conference — Submitted to ICLR 2024_

### Official Review · Reviewer_HcSy · 2023-10-24

**Soundness:** 3 good
**Presentation:** 3 good
**Contribution:** 2 fair
**Rating:** 6
**Confidence:** 3

**Summary:**

The authors set out to map multimodal networks trained on text and vision to predict intracortical recordings from the brain using sEEG for epilepsy during movie watching. They find clear instances of sites encoding both language and text at the temporoparietal junction.

**Strengths:**

This is an informative and well-done study of multimodality measured through intracortical recordings. The data is unique and abundant and the evaluation was done for a large number of models. A lot of attention was put into the controls. As someone who specializes in the field of task-driven neural networks vs brains, I can appreciate that this is well-executed and will surely find a receptive audience for neuroscientists.

**Weaknesses:**

I don't think this is an appropriate venue for the paper. There's no clear methodological advance in ML that would be of broad interest to the ICLR community: it needs to be read by neuroscientists, not ML people. I looked at the neuroscience papers at ICLR in the last 2 years and found only one that would classify as investigating task-driven neural networks in brains in the style of Yamins, DiCarlo, Kriegeskorte, etc., and that paper showed a clear methodological contribution (https://openreview.net/forum?id=Tp7kI90Htd). The authors should look at where this type of neuroAI work is typically published, e.g. NeurIPS, SVRHM, cosyne, PNAS, Nature Comms, etc.

Edit Nov 23rd: I have no real qualms about what's presented in this paper; in response to my original comments, the authors have argued that every once in a while, a task-driven neural network paper in this style is published at ICLR. Thus, my original questions about the suitability of this work for ICLR notwithstanding (and I will note, consistent with reviewer JsYg), I have increased my score to a 6.

**Questions:**

-

**Details Of Ethics Concerns:**

-

---

> ### Author Response · Authors · 2023-11-23
> **Response to reviewer HcSy (1/1)**
>
> Thank you for your review.
> >I don't think this is an appropriate venue for the paper. There's no clear methodological advance in ML that would be of broad interest to the ICLR community: it needs to be read by neuroscientists, not ML people. I looked at the neuroscience papers at ICLR in the last 2 years and found only one that would classify as investigating task-driven neural networks in brains in the style of Yamins, DiCarlo, Kriegeskorte, etc., and that paper showed a clear methodological contribution (https://openreview.net/forum?id=Tp7kI90Htd). The authors should look at where this type of neuroAI work is typically published, e.g. NeurIPS, SVRHM, cosyne, PNAS, Nature Comms, etc.
>
> First, we would like to mention that task-driven neural networks in brains have been investigated, even last year in ICLR 2023 (https://openreview.net/forum?id=KzkLAE49H9b). Furthermore, papers that have non-ML takeaways have been accepted to ICLR, both with the paper we linked and other papers in the neuroscience and cognitive science category(https://openreview.net/forum?id=xmcYx_reUn6 ).
>
> We believe this paper could be of great interest to the ICLR and neuroscience community and has a strong methodological contribution for any person interested in understanding how well their neural network computation matches computation in the brain. For the neuroscience community, the obvious takeaway is the identified brain region. For the greater deep learning community, there is a greater interest in connecting DNNs with the brain. Specifically,
>
> 1. There are numerous attempts, like BrainScore, in neuroscience and ML to make models more brain-like so as to have the robustness of humans. By showing a link between multimodal models and the brain, we can extend this search to multimodal architectures.
>
> 2. Traditionally, the ML community has had one objective: maximize benchmark performance. Recent language models and multimodal models excel at existing benchmarks making progress increasingly difficult. We enable comparisons against the brain, providing a useful new objective function and a specific statistical methodology to analyze the objective function on our data.
>
>
> 3. Combining 1 and 2, at the next ICLR, you could see a modified multimodal Transformer that is far more similar to the brain. This would not be possible without our work, both in establishing what exists now and laying down a coherent methodology for doing this research.
>
> 4. The novelty of our methodology is in the “Confidence Intervals across Time” and “Model Comparisons” sections. Had we carried out this work with the standard analysis, like in Goldstein (2021), ignoring the fine-grained statistical controls we added, we would easily have found numerous multimodal areas, but they would not have been statistically meaningful. There was little need for prior papers to address this methodology because fMRI does not have high temporal resolution and Goldstein (2021) didn’t have the same kind of model comparisons.
>
> 5. Practically, this also shows that multimodal Transformers are likely on to something important. A line of evidence for their architecture that is totally divorced from any engineering benchmark (we as a community design benchmarks so that our models have some hope of handling them, so this line of evidence is somewhat contaminated) and relies only on comparing them to the brain (which we cannot influence as a community).
>
> There are many potential takeaways from our analysis beyond the presentation of regions in the brain that we can list that may be of stronger interest to a deep learning audience in the ICLR community. For example, we could use our comparisons to identify the current best multimodal model across all multimodal electrodes. We include this as a new subsection in Section 4.3 and a new figure.

---

### Official Review · Reviewer_KL3N · 2023-10-31

**Soundness:** 1 poor
**Presentation:** 1 poor
**Contribution:** 1 poor
**Rating:** 3
**Confidence:** 4

**Summary:**

The current paper described a comparison between language/vision unimodal and language-vision multimodal models in predicting brain activities while the subject is watching movies. The authors trained model features to predict SEEG brain activities using ridge regression. Following the performance of the ridge regression on hold-out set of data, the authors identified the brain areas that best fit by a unimodal or multimodal model. This study shed light on the function of cortical areas.

**Strengths:**

Using STOA vision and language models to help interpret the function of the brain is interesting.

**Weaknesses:**

Overall, no insights have been generated in the current study, nor have a solid novel methodology.

1.	Missing references in the related work section (paragraph 1, last two sentences; paragraph 2, 1st sentence). Missing figure references throughout the text. The writing of the paper lacks clarity.

2.	Model performance seems to be low. Figure 2a suggests that the Pearson correlation between model predictions and the true neuron activity is about 0.1. This means the model explains about 1% of the variance of the data (linear model r2). This is pretty low. Please explain why we should care about a model with limited prediction power.

3.	The performance difference between brain regions, or between models is low, compared to the error bar per condition (figure 2a, and in multimodality tests). By the way, it is not clear to me what the error bar stands for. Is the difference between models or between electrodes sufficient for discrimination?

4.	Whether the model selection is consistent with the known biological function of the brain areas.

5.	Do all multimodal tests identify the same set of multimodal selection electrodes? Quantitative results should be provided. A standard procedure should be followed to identify brain areas as potential vision-language integration regions.

6.	How the regression model is trained? What is the input and what is the output? For each model with a different feature size, how does the parameter space for the ridge regression model differ between models? Whether the model performance was affected by the number of parameters?

**Questions:**

1.	Clarify the analysis and model comparison criteria.
Figure 1a suggests that the Pearson correlation coefficient is obtained per time bin per electrode per model. What exactly is in the vector that feeds into the Pearson correlation analysis?
How many image-text pairs are in the training, testing, and validation dataset? What is the fraction that achieved above threshold prediction?
Provide example predictions and the corresponding true brain activities, and provide the Pearson r value for the example.
2.	Improve figure resolution, please.
3.	Address all the questions raised in the Weakness section.

---

> ### Author Response · Authors · 2023-11-23
> **Response to reviewer KL3N (1/1)**
>
> Thank you for your thoughtful review. We respond to the points below.
> >Model performance seems to be low. Figure 2a suggests that the Pearson correlation between model predictions and the true neuron activity is about 0.1. This means the model explains about 1% of the variance of the data (linear model r2). This is pretty low. Please explain why we should care about a model with limited prediction power.
>
> Note that Figure 2a averages the Pearson correlation over many models, neurons (electrodes) and time bins so this is not entirely representative of the meaningful range of possible Pearson r values. Recall that our window size extends from -2s to 2s and predicting activity in a time bin close to the tail end of this window size gets very low Pearson correlation. We are including these in the average in order to have a fuller representation in the plot. As stated in Section 4, we can obtain max Pearson correlation values of ~0.52 for particular time bins and neurons.
>
> >The performance difference between brain regions, or between models is low, compared to the error bar per condition (figure 2a, and in multimodality tests). By the way, it is not clear to me what the error bar stands for. Is the difference between models or between electrodes sufficient for discrimination?
>
> Apologies for the confusion. We should have a better caption for Figure 2, which now has been moved to Figure 7 in the appendix. Once again, the error bar in Figure 2a is generated from averaging the raw Pearson correlation scores from running a single base regression. All scores in section 4c refer to the intervals that we obtain from bootstrapping analyses so this range will be different. Based on the differences we report in this section, we believe that the difference between multimodal and unimodal models is significant. Multimodal models outperform unimodal models by 0.07 points on the language-aligned dataset and 0.09 on the vision-aligned dataset.
>
> >Whether the model selection is consistent with the known biological function of the brain areas.
>
> For selecting unimodal models, we selected models with general architectures that were used in prior work extensively such as vision transformers (SLIP-SimCLR, BEIT), CNNs (ConvNeXt), or language transformers (SBERT, SimCSE). These models have been shown to fit the brain consistently. For our multimodal selection, we stayed with the general motif of transformer architectures which generally have seen success at modeling activity in the brain. We selected the best-performing SOTA DNN models of vision-language processing at the time. Connecting DNN computations and circuitry to neural computations and circuitry is still an open question but newer work has shown connections between the transformer architecture and the brain.
>
> >Do all multimodal tests identify the same set of multimodal selection electrodes? Quantitative results should be provided. A standard procedure should be followed to identify brain areas as potential vision-language integration regions.
>
> Our dataset-specific tests of multimodality (test 1 and 3) do identify potentially different electrodes which is why we take an intersection between electrodes that pass test 1 for the language-aligned dataset and vision-aligned dataset for test 2 (and do the same between test 3 and test 4). We are a bit unclear on how this isn’t a standard procedure or what quantitative results the reviewer wants to see. We believe we are using a standard procedure (bootstrapping model performance difference) as a way of comparing models over the significant time bins -- this is a general procedure.
>
> >How the regression model is trained? What is the input and what is the output? For each model with a different feature size, how does the parameter space for the ridge regression model differ between models? Whether the model performance was affected by the number of parameters?
>
> Sorry for the confusion! We do discuss these details in Appendix C -- due to space this discussion can’t find entirely in the main paper. To summarize, the regression model is trained as follows. We have a set of image-text pairs, $(i, t)$. We feed the pair (or one input in the tuple) to our DNN model, $M$, to get $r = M(i, t)$, where r is the activations of the DNN in response to $(i, t)$. We also extract a 4s window of sEEG activity from one electrode e and split this into 161 time bins of averaged activity, $e_t$. The regression is trained to take $r$ as input and predict $e_t$ for each time bin across all electrodes. So, the number of parameters in the regression is dependent on the dimensionality of $r$ which is affected by the dimensionality reduction procedure. We found that the number of parameters in the regression did not affect the regression performance across models.

---

### Official Review · Reviewer_gGN5 · 2023-11-02

**Soundness:** 2 fair
**Presentation:** 3 good
**Contribution:** 3 good
**Rating:** 6
**Confidence:** 5

**Summary:**

There is a multitude of papers training decoders from latent representations of DNN models onto brain activity. The goal is to reconstruct the brain activity, which, if successful, would indicate that there is something brain-like in the representations that are formed by artificial learning models. One cohort of such papers focus on models of vision (and predicting activity of visual areas of the human brain), while another on language models (and predicting the activity of language areas in the brain). In this paper the authors ask what if we take models that work on vision and language simultaneously - would the representations and activations of those models be more predictive of brain activity? And if so - in which regions?

An answer to this question might help us understand where in the brain are the areas that integrate different sensory modalities together, or at least work on several modalities at the same time.

The authors then take 14 models and use their representation to predict each electrode's recordings. The idea is that if a multimodal model's representation is significantly more useful for predicting the brain activity, then this model's representations are closer to what is happening in the brain, and thus can be thought of as evidence for that electrode's area being involved in multimodal processing of information.

The authors do find several such electrodes, but the number of those electrodes is not sufficiently high to draw strong conclusions (at least this was my impression from reading the paper, please correct me if I am wrong).

Overall this work presents and very cool idea, which is well-executed and logically reported, however due to the lack of data (I suspect) there is just not enough ground to claim definitive findings.

**Strengths:**

Indeed an important and timely question, and, to the best of my knowledge, this work is first to explore this topic. I really like the idea and the research question.

I very much support the emphasis you made on the fact that before any analysis is done we should confirm that there is either a strong difference in signal reconstructive power between trained and untrained networks, or a strong decoding ability -- we need to have a confirmation that the signal is indeed there before we analyse anything related to it.

**Weaknesses:**

My overall impression can be summarised as follows: while the first half of the paper is great and explains the idea and motivation really well, creating a rightful sense of expectation of the result, the section on the results somewhat comes short of delivering the findings with a bang. After reading the first half I was excited to read the next pages to find out "where, indeed, are those areas that integrate vision and language?" and tingling with an expectation of learning something new about our brains. But then, for some reason, the Results section is very timid and just presents dry numbers for each of the test that were planned. Were the differences in Pearson R not strong enough for the authors to be confident in their findings? What would be the strong result? Is this just the matter of presentation, or the result is too weak to claim some sort of victory and knowledge discovered?

(1) It would be helpful to include a better explanation of what the "event structures" are exactly. Maybe a picture.

(2) Page 6, Section 4, second paragraph: This bit of text here is a bit too overloaded with number and while the authors might expect their readers to be careful and try to understand what is the meaning and significance of those numbers being what they are... but a reader is rarely that careful. I would advise to add explanations to this paragraph that explain to the reader what they are supposed to think when they see this or other set of numbers. Tell the reader what they are supposed to with those numbers.

(3) Figure 2: The caption does not explain the figure well. Panels (a) and (b) are not mentioned in the caption. An attempt to explain with "mid left" and "bottom right" is confusing, perhaps just put the names of the models on the figure plots. The dots on the freesurfer brain surfaces on the right of the figure are not explained at all - what are they? For the colour-blind it is very hard to see the red dot, I recommend using blue.

(4) Page 8, Section 4.2, second test paragraph: It is unclear without further explanations what do the authors themselves make of this result. Was it interesting and/or significant? What did it demonstrate? Was this a strong result or no so much? Expected or unexpected? All in terms of the main research question of the paper.

(5) Same as (5) applies to paragraphs on tests three, four and five. Actually one also. Currently these paragraphs are basically just a table of results but written out in words. A table with results and numbers is good, but we also need an interpretation and analysis of these results. Are they strong / interesting? Is there a scientific discovery here? What is it? How strong is the evidence? Were the R differences significant?

(6) Page 9, first paragraph: "we find that the largest contiguous cluster of these electrodes is found in and around the temporoparietal junction" -- it would help a lot to see those electrodes on a picture of a brain! Not only in a way how it was presented on Figure 4, but actually each electrode plotted as a dot so that the reader could also discover this by actually seeing that "yep, indeed, here are the electodes that are more predictive in multimodal regime and indeed they cluster around superior temporal region". Figure 4, in my opinion, falls short of presenting this finding and sweeps the results under average-colored areas, raising questions why the plot was individual electrodes was not shown. Something like Figure 8 (supplementary G), but please use different colors for "unimodal" or "multimodel across vision and language" as the current selection of colors blurs together

(7) In your own estimation, are those singular electrodes shown on Figure 8 as "multimodal across vision and language" provide sufficient evidence to multimodal processing in those areas? Or are the numbers too few to provide strong support for this claim? The yellow dots seem to be quite scattered, and we also don't know, for example, is the lack of them in visual areas explained by the fact that no electrodes in those areas were multimodal, or is this just because there were not electrodes implanted there?

(8) It would help to evaluate the strength of the funding if we would be able to see a comparison (maybe a distribution plot) of multimodal over language-only / vision-only -- this would allow us to see not only where and how much of those electrodes exist, but also _how different_ their predictive power is. The averaged numbers you provide in section 4 are just averages and are just one number, hiding the true distribution we would be interested to see.

**Questions:**

(1) The implantation sites of sEEG electrodes in your dataset were clinically motivated. To what extent did they cover the areas you were interested in? Both whether all of the areas of interest were covered, and also among the the areas that were covered - was the coverage sufficient for you analysis in your estimation and why?

(2) Are there multimodal models combining vision, text and audio? The data that you have contains all 3 modalities, what was/is the main obstacle to identifying tri-modal predictive regions in the brain? Is it the lack of appropriate DNNs or something else?

(3) In your experiments were the subjects able to hear the audio track of the movie?

(4) For your literature review here is another work comparing vision to DNN specifically on sEEG data from a 100+ subjects https://www.nature.com/articles/s42003-018-0110-y

(5) How do you deal with the fact that the data comes from different subjects? Does inter/intra-subject considerations enter your analysis at all or you just consider each LFP electrode on its own regardless of the subject it came from?

(6) Could you perhaps use fMRI data instead? You would lose temporal and frequency resolution, but for the level of analysis at which you are working these are not too relevant and demonstrating higher predictability of BOLD signal would be equally impressive and informative. More importantly it would allow you to capture the whole brain and there should be more datasets available for fMRI that for sEEG.

---

> ### Author Response · Authors · 2023-11-23
> **Response to reviewer gGN5 (1/2)**
>
> Thank you for your thoughtful and detailed review! We address your points below.
>
> **Weaknesses:**
>
> >(1) It would be helpful to include a better explanation of what the "event structures" are exactly. Maybe a picture.
>
> Apologies, we realize the concept is a bit confusing. We include a figure in the appendix showing event structures selection in Appendix A.2.
>
> >(2) Page 6, Section 4, second paragraph: This bit of text here is a bit too overloaded with number and while the authors might expect their readers to be careful and try to understand what is the meaning and significance of those numbers being what they are... but a reader is rarely that careful. I would advise to add explanations to this paragraph that explain to the reader what they are supposed to think when they see this or other set of numbers. Tell the reader what they are supposed to with those numbers.
>
> Thank you! We have updated the results section to better address the numbers here. Our goal was to present the results without overselling/overclaiming our findings but we realize that our current writing may come off as dry or needlessly confusing to a reader.
>
> >(3) Figure 2: The caption does not explain the figure well. Panels (a) and (b) are not mentioned in the caption. An attempt to explain with "mid left" and "bottom right" is confusing, perhaps just put the names of the models on the figure plots. The dots on the freesurfer brain surfaces on the right of the figure are not explained at all - what are they? For the colour-blind it is very hard to see the red dot, I recommend using blue.
>
> Thank you! We have decided to focus on some new results and have moved Figure 2 to the appendix as Figure 7. In addition, we have updated the caption and figure.
>
> >it would help a lot to see those electrodes on a picture of a brain!
>
> This makes sense! We now include Figure 4 which includes a visualization of the most predictive multimodal model on all of the multimodal electrodes. We hope this better addresses this comment.
>
> >(7) In your own estimation, are those singular electrodes shown on Figure 8 as "multimodal across vision and language" provide sufficient evidence to multimodal processing in those areas?
>
> The improvement in Pearson’s r that we achieve is significant. We introduce many controls on the dataset, architecture, and statistics. We still achieve significant gains from multimodal models, sometimes close to r=0.1. We agree with the reviewer that our current draft may leave the reader feeling confused as to whether our results are trustworthy so we include some new text focusing on this.
> In general, we approach these results with caution due to the fact that little is known about multimodal integration, especially with vision and language. We are excited to see our results match prior work. We are also excited to see that our results are not explained by better task performance. We believe that this is a strong start to deeper exploration into these areas and what in our networks drives our prediction. That being said, our motivation for showing numbers was not to deter readers or reduce confidence but instead to be factual. We believe these studies are invaluable given the lack of understanding around multimodal integration.
>
> **Questions:**
>
> >(1) The implantation sites of sEEG electrodes in your dataset were clinically motivated. To what extent did they cover the areas you were interested in? Both whether all of the areas of interest were covered, and also among the the areas that were covered - was the coverage sufficient for you analysis in your estimation and why?
>
> Great question, and this likely connects with the weakness you were bringing up. There were some areas that were not covered in the implantation such as early visual cortex. These are areas where we know significant visual processing occurs but these could not be included in our analyses. We also had fewer electrodes in the occipitotemporal cortex or the superior frontal cortex than we would have liked, making the percentage of multimodal electrodes small in these regions as shown in Figure 4.
>
> Figure 5 shows that we have broad coverage across the brain but a majority of the electrodes naturally fell into the superior temporal and middle temporal cortex -- one subject had all of their electrodes in this location. We had coverage across all desirable locations of the brain other than EVC and having more coverage in OTC or the superior frontal cortex would have given more evidence for these regions to be multimodal.

---

> > ### Author Response · Authors · 2023-11-23
> > **Response to reviewer gGN5 (2/2)**
> >
> > >(2) Are there multimodal models combining vision, text and audio? The data that you have contains all 3 modalities, what was/is the main obstacle to identifying tri-modal predictive regions in the brain? Is it the lack of appropriate DNNs or something else?
> > f
> > There are [1]! The main obstacle for this paper is processing time -- none of our methods prevent us from extending to one more modality. Obviously, we would need to increase the number of baselines to have a meaningful comparison which would be difficult in the time we have for this rebuttal. We would like to have this as future work where we compare 3 modalities vs 2 modalities vs 1 modality with these models. We would also like to include video models as a comparison as well.
> >
> > >(3) In your experiments were the subjects able to hear the audio track of the movie?
> >
> > Yes, subjects were able to hear the audio track. The movie presentation was unaltered from a normal movie viewing experience.
> >
> > >(4) For your literature review here is another work comparing vision to DNN specifically on sEEG data from a 100+ subjects https://www.nature.com/articles/s42003-018-0110-y
> >
> > Thank you! We will add this to our related work section.
> >
> > >(5) How do you deal with the fact that the data comes from different subjects? Does inter/intra-subject considerations enter your analysis at all or you just consider each LFP electrode on its own regardless of the subject it came from?
> >
> > Yes! We consider each electrode separately and use the data from the movie the subject viewed to test how well a DNN modeled activity. We didn’t do any inter/intra-subject analysis other than in the final steps where we pooled the electrodes together to find general regions from the DKT atlas.
> >
> > >(6) Could you perhaps use fMRI data instead? You would lose temporal and frequency resolution, but for the level of analysis at which you are working these are not too relevant and demonstrating higher predictability of BOLD signal would be equally impressive and informative. More importantly it would allow you to capture the whole brain and there should be more datasets available for fMRI that for sEEG.
> >
> > Our analyses certainly would apply to fMRI data! We are unaware of fMRI datasets that use a rich multimodal stimulus similar to movies. We feel that sEEG allows us to go deeper in future work though, e.g. we can follow this paper up with temporal analyses which are not possible in fMRI. We recognize that it would be worthwhile to obtain similar analyses with neural imaging that has more coverage than sEEG and we can look into fMRI analyses in the future.
> >
> > [1] Shvetsova et. al. Everything at Once -- Multi-modal Fusion Transformer for Video Retrieval. CVPR 2022. https://arxiv.org/abs/2112.04446

---

### Official Review · Reviewer_9iyA · 2023-11-02

**Soundness:** 3 good
**Presentation:** 3 good
**Contribution:** 3 good
**Rating:** 6
**Confidence:** 4

**Summary:**

This paper aims to identify neural sites where multimodal integration is occuring in the brain. To achieve that, authors evaluate in which regions  multimodal (vision + language) models are better than unimodal models in predicting neural recordings (SEEG).

Using this method, the authors identify 141 out of 1090 total sites where multimodal integration is happening.

**Strengths:**

1. The methodology to identify multimodal sites is well described and quite comprehensive (trained vs. random, multimodal vs. unimodal, SLIP combo vs SLIP-SimCLR). The release of code will enable future work investigating similar questions with other modalities or other type of brain recordings
2. The paper is easy to follow. This is due to clear writing and presentation of methods and results.
3. Statistical tests and confidence intervals.
4. Multimodality test results on section 4.2. I appreciate 5 tests of multimodality reported in the results and how each test  filters out possible confounds.

**Weaknesses:**

1. One test that can also be included is to randomly input one of the modalities in a multimodal model and then compare with predictions using actual multimodal inputs. This test can reveal the importance of multimodal information avoiding the confounds due to architecture, parameters, training set etc.
2. Did the authors perform SLIP-CLIP vs SLIP-SimCLR vs SLIP-Combo comparison? Because SLIP-CLIP is also multimodal I am curious what was the motivation for only showing SLIP-CLIP vs Combo results in Figure 4
3. It is not clear to me why language alignment and vision alignment event structures leads to difference in results. I would like to read author’s explanation on why results depend on how event structures and if this is a limitation of this approach.

**Questions:**

1. Is the dataset also part of this paper or is it from an already published paper? If yes then this is an additional contribution of this paper and should be emphasized.
2. Will this dataset be publicly released? ( if not released already )

---

> ### Author Response · Authors · 2023-11-23
> **Response to reviewer 9iyA (1/2)**
>
> Thank you for your thoughtful review. We address your suggestions below.
>
>
> >One test that can also be included is to randomly input one of the modalities in a multimodal model and then compare with predictions using actual multimodal inputs. This test can reveal the importance of multimodal information avoiding the confounds due to architecture, parameters, training set, etc.
>
> Thank you for this suggestion! Due to the time constraints and limited resources, we could only perform this permutation for 1 multimodal electrode and one model. This test only applies to architecturally multimodal models such as BLIP, which we use in this test.
>
> More specifically, we randomly permuted one modality in our event structures 1000 times and fed the new inputs to BLIP and predicted activity in the 1 multimodal electrode. This led to four types of permutations: language-aligned dataset where we permute the language modality, language-aligned dataset where we permute the vision modality, vision-aligned dataset where we permute the language modality, and vision-aligned dataset where we permute the language modality. We calculate the p-value for each comparison in this permutation test.
> We find that for the language-aligned dataset, when we permute either modality, we obtain a p-value of 0.001 (meaning none of the 1000 scores is larger than the unpermuted score). For the vision-aligned dataset, our permutation passes (p-value < 0.05) when we permute the vision modality but not when we permute the language modality (p-value = 0.169). We have two reasons for this. For the language-aligned dataset, BLIP is the best model that explains activity in the electrode. In the vision-aligned dataset, the best model is SLIP-Combo. In general, architecturally multimodal models are worse at modeling vision-language integration than the trained multimodal models like SLIP-CLIP or SLIP-Combo. Furthermore, the alignment presented in the vision-aligned dataset is weaker. Language is not consistently paired with the main visual stimuli, scene-cuts. Our architecturally multimodal models may not be good enough to model this form of integration. We hope to run a larger version of this test but the computation is expensive and requires us to reload embeddings 1000 times for every subject and run 1000 regressions across all electrodes. This test may be better with newer architecturally multimodal models such as BLIP-2.
>
> >Did the authors perform SLIP-CLIP vs SLIP-SimCLR vs SLIP-Combo comparison? Because SLIP-CLIP is also multimodal I am curious what was the motivation for only showing SLIP-CLIP vs Combo results in Figure 4.
>
> This is a good point! We could have included SLIP-CLIP as well but chose SLIP-Combo and SLIP-SimCLR as our main comparison. The choice between SLIP-CLIP and SLIP-Combo as our multimodal choice was arbitrary. We show the results between SLIP-CLIP and SLIP-SimCLR for comparison in Appendix H.
>
> >It is not clear to me why language alignment and vision alignment event structures leads to difference in results. I would like to read author’s explanation on why results depend on how event structures and if this is a limitation of this approach.
>
> This is an important point and we add a few sentences to Appendix A.2. Event structures are a first-order approximation for choosing image-text pairs without imposing a strong hypothesis on the structure of real vision-language integration. Vision-aligned data selects scene cuts and the closest sentence and this may be modeling longer-term integration of vision and language than the language-aligned dataset which uses context segments and frames. This is because the vision-aligned data is less aligned than the language-aligned data since a frame occurs with every word but a sentence can occur much later (sometimes, 2 seconds) after a scene cut.
> We believe the event structure selection has strengths and limitations. Our event structures are naturalistic and intuitive, prioritizing one modality and sampling the other modality. They do not rely on model architecture or make assumptions about vision-language integration. They also cover many forms of possible integration by strictly aligning (language-aligned) and loosely aligning (vision-aligned) the modalities. However, we recognize that in an ideal world, we would have defined vision-language events similar to words or objects that are easily extractable from our movie dataset where we have reasonable evidence that vision-language integration is occurring. In the end though, we argue that the event structure selection makes for a more difficult benchmark by greatly expanding the potential forms of vision-language integration that multimodal architectures need to consider.

---

> > ### Author Response · Authors · 2023-11-23
> > **Response to reviewer 9iyA (2/2)**
> >
> > >Is the dataset also part of this paper or is it from an already published paper? If yes then this is an additional contribution of this paper and should be emphasized.
> >
> > The neural recordings in this dataset are unpublished. We plan to release the dataset of 7 neural recordings from the 7 subjects when making this work public. However, for the purposes of this paper, we don’t plan to emphasize the dataset since this will be under review in a different submission as a dataset paper.

---

> > > ### Comment · Reviewer_9iyA · 2023-11-23
> > >
> > > Thanks for adding new results and explaining the event structure. I have no more questions and will keep my rating.

---

### Official Review · Reviewer_JsYq · 2023-11-03

**Soundness:** 2 fair
**Presentation:** 2 fair
**Contribution:** 2 fair
**Rating:** 3
**Confidence:** 5

**Summary:**

I have reviewed this work previously and the current version does not address my previous major concerns so I will repeat my points in this review in hopes that the authors can address them this time.

This work investigates the ability of multi-modal neural networks to align with multi-modal ECoG brain recordings, acquired while 7 epileptic children were watching movies. The authors aim to use a contrast between multi-modal and uni-modal models to reveal which locations in the brain relate to integration from multiple modalities. A large number of models are tested (7 multi-modal and 5 uni-modal). This work finds that about 13% of the tested neural sites are better predicted by multi-modal models.

**Strengths:**

- Using multi-modal brain recordings from movies
- Using ECoG for high spatial and temporal precision
- Checking many models (12, 2 models in depth)
- An additional analysis that goes more in depth because, as the authors realize, a difference in brain predictivity in a direct comparison between a multimodal and a unimodal model can be due to the many possible differences between the models

**Weaknesses:**

1. The biggest weakness is the central claim that the proposed analyses of multi-modal models can localize vision-language **integration** in the brain. Can the authors please define what they mean by vision-language integration? Even on the model side, it is an open question to what degree multi-modal models actually integrate information from multiple modalities as opposed to increasing the alignment between individual modalities (Liang et al. 2022 https://arxiv.org/pdf/2209.03430.pdf). It is not clear whether the results that are observed are due to vision-language integration or whether they are due to improved representations of the language-only or vision-only modality. For example, in Fig 4, all regions that are identified as multimodal (e.g. marked with a green star), are canonical language regions. How can the authors disentangle the effect of integration of modalities vs the effect of improving language-only information in the model? For instance, let's do a thought experiment and apply the authors' methods to study different layers of a language-only encoder: I predict that these results will be similar to what has been shown before which is that some layers in the language model predict exactly the same regions they call multimodal substantially better than other layers (see Jain and Huth, 2018 NeurIPS; Toneva and Wehbe, 2019 NeurIPS for some of the earlier work showing this). That clearly is not due to multimodality though because the input is only language.

2. The presentation can be much improved: there is very little discussion of what the different models that are used are and how they are trained. This is key to understand the contributions of this work. I suggest the authors include a table of all models and model variations used with clearly marked information about what modality was used to train the model and what modality is used to evaluate the model (e.g. even if the authors are using only a vision encoder at inference time, if the vision encoder was jointly trained with a language encoder, this should be noted as this may make a difference in the representations)
3. The work is still not positioned well in the current literature on multi-modal modeling of brain recordings, and it’s not clear what the novelty here is for people who are unfamiliar with this area. The authors should discuss the work of Oota et al. 2022 COLING and Wang et al. 2022 bioRxiv https://www.biorxiv.org/content/10.1101/2022.09.27.508760v1.
4. The contribution to an ML audience is not very clear. I believe this work will be better suited to be evaluated by neuroscientists, since the claimed contributions are on the neuroscience side, and will also be more appreciated at a neuroscience venue.

**Questions:**

See Weaknesses above

---

> ### Author Response · Authors · 2023-11-23
> **Response to reviewer JsYq (1/3)**
>
> Thank you for the review. We address your points below.
>
> **Weakness 1**: We address this point-by-point below.
>
> >The biggest weakness is the central claim that the proposed analyses of multi-modal models can localize vision-language integration in the brain. Can the authors please define what they mean by vision-language integration?
>
> This is a good point. We define vision-language integration to be any non-linear combination of vision-language features as carried out by the candidate neural networks. Our candidate DNN models like ALBEF, Flava, BLIP, SLIP-Combo etc. represent a hypothetical integration of vision-language features. Effectively, we are testing whether these DNNs model activity in the brain better than language or vision models.
>
> Using neural networks in this way is desirable. We are not attaching ourselves to a specific hypothesis of vision-language integration but instead allowing us to have a larger search space.
>
>
> >Even on the model side, it is an open question to what degree multi-modal models actually integrate information from multiple modalities as opposed to increasing the alignment between individual modalities (Liang et al. 2022 https://arxiv.org/pdf/2209.03430.pdf). It is not clear whether the results that are observed are due to vision-language integration or whether they are due to improved representations of the language-only or vision-only modality.
>
> Could the reviewer explain what they mean by “increasing the alignment between individual modalities”? How is this distinct from integration? The citation the reviewer provides also does not seem to provide any evidence for their claim as we understand it -- Section 4 gives evidence that models perform some alignment between modalities that allows for improvement on many tasks that require some form of multimodal reasoning such as visual question answering, etc. The paper doesn’t show that better unimodal representations are necessarily what leads to improved performance on tasks VQA -- if this was true, architectures like LXMERT or VisualBERT would still be SOTA as long as we introduce better visual or language representations which is not the insights that led to models like BLIP, ALBEF or Flava. Furthermore, we are unsure how our results don’t dispel this doubt. We show that multimodal models significantly out-perform both vision and language models at predicting vision- and language-aligned data. We show that these models do not have strictly better language or visual reasoning, even on our dataset in Table 5 and Table 6. We include a combined version of Table 5 and 6 which presents average language and vision task performance and a new subsection discussing this in Section 4.3. We also include Tables 5 and 6 below.
>
> | Model                       | Test Set Perplexity (One Movie) |
> |-----------------------------|---------------------------------|
> | BLIP                        | 168.7                          |
> | ALBEF                       | 223.8                        |
> | Flava                       | 202.2                           |
> | SLIP-Combo Language Encoder | 197.6      |
> | SLIP-CLIP Language Encoder  | 259.3  |
> | SBERT                       | 121.3                        |
> | SimCSE                      | 145.4                       |
>
>
> | Model                      | Scene Classification Accuracy (Places365) |
> |----------------------------|-------------------------------------------|
> | SLIP-Combo Vision Encoder  | 73.9                                      |
> | SLIP-CLIP Vision Encoder   | 71.2                                      |
> | SLIP-SimCLR Vision Encoder | 71.0                                      |
> | BEIT                       | 75.4                                      |
> | ConvNeXt                   | 76.2                                      |
> | ALBEF                      | 60.1                                      |
> | Flava                      | 58.4                                      |
> | BLIP                       | 62.4                                      |

---

> > ### Author Response · Authors · 2023-11-23
> > **Response to reviewer JsYq (2/3)**
> >
> > >For example, in Fig 4, all regions that are identified as multimodal (e.g. marked with a green star), are canonical language regions. How can the authors disentangle the effect of integration of modalities vs the effect of improving language-only information in the model? For instance, let's do a thought experiment and apply the authors' methods to study different layers of a language-only encoder: I predict that these results will be similar to what has been shown before which is that some layers in the language model predict exactly the same regions they call multimodal substantially better than other layers (see Jain and Huth, 2018 NeurIPS; Toneva and Wehbe, 2019 NeurIPS for some of the earlier work showing this). That clearly is not due to multimodality though because the input is only language.
> >
> > Our experiment shows that multimodal features can predict regions substantially better than unimodal language or vision features across two dataset constructions which prioritize either modality. We demonstrate that our multimodal models cannot have better unimodal representations or reasoning in Table 5 and Table 6 of the appendix as well as the other tables which all show that the multimodal models we choose do not have the same visual or language processing power as our unimodal models. We emphasize that we give our language and vision encoders the best chance to succeed. We choose the best performing layer in the prediction and see considerably high scores. Despite having worse unimodal representations according to the task performance and doing the strongest possible selection to allow unimodal models to out-perform multimodal models, this is not what we see. The thought experiment the reviewer proposes is orthogonal.ayer comparisons in language models can reveal that particular layers are better at predicting these areas than others. This wouldn’t be surprising either; prior work shows language models can predict OTC activity, generally associated with vision processing, reasonably well. Had we just presented a comparison between multimodal and unimodal models, we would agree with the reviewer’s claim but this is not what we do.
> >
> > We acknowledge that the readers may want to see Table 5 and 6 in the main paper and for us to move that discussion out of the appendix. We add a new subsection focused on this discussion to the results section.
> >
> > **Weakness 2**:
> >
> > >The presentation can be much improved: there is very little discussion of what the different models that are used are and how they are trained. This is key to understand the contributions of this work. I suggest the authors include a table of all models and model variations used with clearly marked information about what modality was used to train the model and what modality is used to evaluate the model (e.g. even if the authors are using only a vision encoder at inference time, if the vision encoder was jointly trained with a language encoder, this should be noted as this may make a difference in the representations)
> >
> > We do include a longer discussion in appendix B of the paper. We can make this into a table as well which we include in Appendix B. If the reviewer is still unsatisfied with the discussion, specific points on what remains confusing would be much appreciated.
> >
> > **Weakness 3**:
> >
> > >The work is still not positioned well in the current literature on multi-modal modeling of brain recordings, and it’s not clear what the novelty here is for people who are unfamiliar with this area. The authors should discuss the work of Oota et al. 2022 COLING and Wang et al. 2022 bioRxiv https://www.biorxiv.org/content/10.1101/2022.09.27.508760v1.
> >
> > Both papers are discussed in our current related work section but we expand the discussion as part of a re-worked related work section in the paper to focus more on the multimodal processing and less on fitting models to the brain.

---

> ### Author Response · Authors · 2023-11-23
> **Response to reviewer JsYq (3/3)**
>
> **Weakness 4**:
>
> >The contribution to an ML audience is not very clear. I believe this work will be better suited to be evaluated by neuroscientists, since the claimed contributions are on the neuroscience side, and will also be more appreciated at a neuroscience venue.
>
> We believe that there is a core contribution to the ML audience for comparing neural networks with the brain. We design a standard pipeline with statistical comparisons that lend credibility to a conclusion that one model fits the brain better than another.
> We highlight our ML contributions in an updated subsection 4.4. In Figure 4 we show the best multimodal model across our multimodal electrodes that we find. We believe this investigation can lead to natural follow-up work with our data and pipeline such that in future years, new papers with new multimodal transformers can be released and these architectures are hopefully more similar to the brain. We also want to point out that prior papers have been accepted to ICLR despite not having considerable ML contributions such as [1] or [2]. We believe our work is in-line with what is presented at ICLR.
>
> [1] Wang et. al. BrainBERT: Self-supervised representation learning for intracranial recordings. ICLR 2023.
>
> [2] Loong Aw and Toneva. Training language models to summarize narratives improves brain alignment. ICLR 2023.

---

> ### Comment · Reviewer_JsYq · 2023-12-03
>
> Thanks for the response.
> My main concern still stands. The authors now clarify that what they mean by vision-language integration is "any non-linear combination of vision-language features". However, *any* non-linear combination of vision-language features can also include a mapping in which the vision features are even exactly the same as they were before the non-linear combination but the language features are combined in new non-linear ways. Would the authors call this integration? I would not. This is related to the thought experiment I proposed. If I apply the authors' methods to study different layers of a language-only encoder, which recombines the previous layers in non-linear ways, I will find that some layers predict brain responses better than other layers. But this is clearly not due to any multimodality.
>
> I am keeping my score. I recommend that in an improved manuscript the authors clarify their integration definition and crisply explain how their results support the drawn conclusions. I also recommend clarifying the contributions to an ML audience. The papers that were referred to either have clear methodological contributions (Wang et al.) or interpret ML models (Aw et al.).

---

### Author Response · Authors · 2023-11-23
**General Response (1/1)**

Thank you all for the thoughtful reviews. We’re glad to see reviewers generally found the paper setting interesting (JsYq, 9iYA, gGn5) and saw novelty in our approach (9iYA, gGn5). We have introduced general changes to the paper (which are highlighted in blue) which we cover here:

* Reviewer JsYg points out that our findings could be due to unimodal task performance. We had addressed this in the Appendix of our original draft but now include a new subsection, 4.4, in our paper to address this.
* Reviewer 9iYA, gGN5 asked for more clarification regarding event structures. We add discussion and a new figure to the paper appendix to address this.
* Reviewer JsYg, Hc5y asked about whether our paper would be applicable to the general ICLR community. We add a result about the best performing multimodal model and show that our analyses can give conclusions of interest to ML practitioners.
* Reviewer gGN5 asks for more interpretation of our results.
* Reviewer KL3N, gGn5, Hc5y asked for changes to figure 2. Due to confusion, we moved Figure 2 to the appendix and updated the figure caption.

Our paper is a meaningful step forward in understanding vision-language integration in the brain. We believe this is valuable work for the ICLR community to understand representations in the brain. We will release our code and data to facilitate further research in this line. We highlight all paper changes in blue.

---

### Meta-Review · Area_Chair_PJaU · 2023-12-05

**Metareview:**

This work was found by reviewers to be generally well-executed, well-controlled, in depth (e.g. involved many models and analyses) and statistically satisfactory.

The authors responses and additional work/revisions addressed a number of the *methodical or clarity* concerns raised.

However, overall this paper is not recommended for acceptance to ICLR. As a first cut, numerically the scores fall significantly below what might be regarded as the acceptance threshold or even the borderline zone. More importantly beyond scores -- and this AC considered the reviews and responses in depth and at length -- this work does not make a clear and strong contribution to the ICLR community (even those with neuroscience inclinations, such as myself and most of the reviewers). I elaborate in the following paragraphs.

Multiple reviewers (and this AC) feel that ICLR is not an appropriate venue. The authors point to two previous ICLR papers that are brain-related, specifically involving data from brain recordings; and indeed there are more brain-inspired papers. However, upon closer inspection, the relationship or contribution there is clear. Either the application is analysis of brain data using DL methods, or else the data is used as ground truth to validate DL models. However, that is not the case here. As written in the paper, in this work there is "a lack of ground truth", and the work primarily provides a catalogue of "candidates for future experiments", with unclear or weak contributions to either the ML or neuroscience communities.

Even among some reviewers who gave positive ratings, their reviews are clearly qualified or conditional, e.g. "no insights have been generated", "not enough ground to claim definitive findings".

For a selective/competitive venue like ICLR, this work is not recommended for acceptance in its current form.

**Justification For Why Not Higher Score:**

After careful consideration of the reviewers points and authors responses, despite my also having computational neuroscience background that this work belongs to, I conclude that the work is not of strong value to either ICLR or neuroscience communities in its current form.

Similarly, of the 4 reviewers, those with neuroscience backgrounds (e.g. according to publications in OpenReview profiles) were negative (or positive but in a very conditional/qualified manner) about this work.

**Justification For Why Not Lower Score:**

N/A

---

### Decision · Program_Chairs · 2024-01-16

Reject